# cAMP and c-di-GMP synergistically support biofilm maintenance through the direct interaction of their effectors

Cong Liu [1,2], Di Sun [1,2], Jiawen Liu[1], Ying Chen[1], Xuge Zhou[1], Yunrui Ru[1], Jingrong Zhu[1] & Weijie Liu [1✉]

Nucleotide second messengers, such as cAMP and c-di-GMP, regulate many physiological processes in bacteria, including biofilm formation. There is evidence of cross-talk between pathways mediated by c-di-GMP and those mediated by the cAMP receptor protein (CRP), but the mechanisms are often unclear. Here, we show that cAMP-CRP modulates biofilm maintenance in *Shewanella putrefaciens* not only via its known effects on gene transcription, but also through direct interaction with a putative c-di-GMP effector on the inner membrane, BpfD. Binding of cAMP-CRP to BpfD enhances the known interaction of BpfD with protease BpfG, which prevents proteolytic processing and release of a cell surface-associated adhesin, BpfA, thus contributing to biofilm maintenance. Our results provide evidence of cross-talk between cAMP and c-di-GMP pathways through direct interaction of their effectors, and indicate that cAMP-CRP can play regulatory roles at the post-translational level.

[1] Jiangsu Key Laboratory of Phylogenomics & Comparative Genomics, School of Life Sciences, Jiangsu Normal University, Xuzhou, China. [2]These authors contributed equally: Cong Liu, Di Sun ✉email: leonliu2013@126.com

The cAMP receptor protein (CRP) is a global transcription factor known to control the transcription of numerous genes by responding to changes in the intracellular cAMP level in most bacteria[1]. Extensive genetic, biochemical, biophysical, and structural data have been analyzed to determine how the cAMP-CRP complex controls gene transcription[1–4]. In *Escherichia coli*, apo-CRP (homodimer CRP in the absence of cAMP) is in the "off" state, which binds DNA nonspecifically and weakly[5,6]. On binding cAMP in the N-terminal domain, CRP undergoes an allosteric transition and is activated to the "on" state, which binds DNA specifically and strongly via its C-terminal domain[6]. CRP can directly interact with RNA polymerase or other transcription factors to control the transcription of corresponding genes, and both interactions are promoter-dependent[7–9]. Although known for its role during carbon catabolite repression, the biological role of cAMP-CRP goes far beyond carbon catabolite repression in most bacteria and includes toxin production[10], iron acquisition[11], capsule production[12], and quorum sensing[13]. Thus, the transcription of numerous genes is controlled by cAMP-CRP in bacteria. Indeed, in *E. coli*, the transcription of more than 7% genes is regulated by cAMP-CRP[14,15]. Although most subsequent studies focused on cAMP-CRP as a transcription factor, cAMP-CRP may play a non-transcriptional regulatory role in some bacteria. For instance, in *Mycobacterium tuberculosis*, CRP may act as a nucleoid associated protein (NAP) to influence the dynamic spatial arrangement of the chromosome in a cAMP-independent manner[16–18]. However, in γ-proteobacteria, the researches on cAMP-CRP as a non-transcriptional regulator involved in physiological metabolism are limited.

Biofilms are structured communities of sessile, microbial cells encased in a self-secreted extracellular matrix, which is composed of exopolysaccharides, proteinaceous adhesin factors and nucleic acids[19–21]. The bacterial biofilm developmental process includes four stages: (i) initial attachment, (ii) microcolony formation, (iii) biofilm maturation, and (iv) dispersion[22,23]. There has been an abundance of studies on the regulation of initial attachment in multiple bacteria, less is known about dispersion. To investigate the transition from biofilm maturation to dispersion, the term "biofilm maintenance" has been proposed to describe the process by which existing mature biofilms regulate themselves to persist on a surface before dispersion[24]. The intracellular second messenger bis-(3′-5′)-cyclic dimeric GMP (c-di-GMP) is involved in regulating each stage of biofilm development[24–27]. Diguanylate cyclase (DGC), containing a conserved GGDEF domain, catalyzes two molecules of GTP to synthesize c-di-GMP, which is degraded to 5′-phosphoguanylyl-(3′-5′)-guanosine (pGpG) and/or GMP by phosphodiesterase (PDE) with a conserved EAL/HD-GYP domain[28–30]. Most bacteria have more than one DGC/PDE, the quantity of which is associated with the complexity of the habitat of bacteria[31]. For example, free-living microbes tend to have more DGC/PDE than obligate pathogenic bacteria[31–33]. Although most bacteria have multiple DGCs/PDEs, only a few DGCs/PDEs influence biofilm development at a defined time period[29,32,34–37]. Under most conditions, c-di-GMP is involved in regulating specific regulatory networks by forming a complex with c-di-GMP receptors/effectors[38–41], which switch bacterial transition between sessile biofilm and planktonic modes depending on their ability to bind c-di-GMP[25,37,39]. As a consequence, a high intracellular c-di-GMP level is associated with biofilm formation, while a low intracellular c-di-GMP level tends to facilitate a planktonic lifestyle[31,42,43].

*Shewanella* species are gram-negative facultative anaerobic γ-proteobacteria, which are dissimilatory metal-reducing bacteria and widely distributed in aquatic niches[44,45]. The respiratory diversity and ability to form biofilms of *Shewanella* species allow for its use in various bioremediations and biotechnologies[46–48]. Previous studies have reported that the outer membrane adhesin BpfA promotes the biofilm formation of *Shewanella oneidensis* MR-1 and *Shewanella putrefaciens* CN32 through improving the cell adhesion to a solid surface[49–52]. Moreover, *bpfA* (*Sputcn32_3591*) is the first gene in an operon (Supplementary Fig. 1a) containing seven genes encoding a type I secretion system for the translocation of BpfA protein (Sputcn32_3592, Sputcn32_3593, AggA, Sputcn32_3595), a periplasmic protease BpfG (Sputcn32_3596), and an inner membrane-spanning c-di-GMP effector BpfD (Sputcn32_3597)[49,53,54]. Both BpfG and BpfD control whether BpfA is localized on the cell surface, and together, is defined as the BpfAGD system[55]. The regulation model of the BpfAGD system is similar to the Lap system of *Pseudomonas fluorescens* Pf0-1[42]. As shown in Fig. 1, a high intracellular c-di-GMP level activates the c-di-GMP effector BpfD to bind to and sequester the periplasmic protease BpfG, which prevents BpfA being processed and results in biofilm formation (Fig. 1a). When intracellular c-di-GMP level is low, BpfD cannot sequester BpfG, leaving BpfG free to process and release BpfA from the cell surface, leading to planktonic mode[55].

cAMP-CRP has been reported to regulate biofilm formation in some bacteria. For example, in *Pseudomonas aeruginosa*, a solid surface signal activates two adenylate cyclases, CyaA and CyaB, thereby promoting cAMP synthesis[56]. Subsequently, cAMP activates the transcriptional regulatory activity of Vfr, a homologous protein of CRP, and the Vfr-cAMP complex triggers the transcription of a range of genes that are involved in biofilm formation[57,58]. In addition, as a global transcription factor, cAMP-CRP has been reported to regulate the transcription of some genes encoding c-di-GMP receptors/effectors or DGCs/PDEs to control biofilm formation in some bacteria[59–61]. Compared to c-di-GMP, which acts as a "switch molecule" to control the transition between motile planktonic and sessile biofilm lifestyles in many bacteria, the function of cAMP-CRP in biofilm formation seems more ancillary[62]. Thus, the available research on the underlying mechanisms of cAMP-CRP regulating biofilm formation is limited.

In this work, we show that cAMP-CRP is required to maintain a mature biofilm of *S. putrefaciens* CN32. Further investigations indicate that cAMP-CRP physically interacts with the inner membrane-spanning c-di-GMP effector BpfD, and this interaction greatly enhances the capability of BpfD to interact with and sequester BpfG, thereby retaining BpfA on the cell surface and supporting biofilm maintenance. This report not only reveals that cAMP and c-di-GMP synergistically regulate biofilm maintenance through the direct interaction of their effectors but also describes a regulatory pattern that cAMP-CRP modulates biofilm maintenance acting as a post-translation regulator.

## Results

**cAMP-CRP complex regulates biofilm maintenance**. Genes associated with biofilm maintenance were screened using a transposon insertion mutation technique and *crp* (Sputcn32_0652) was identified. The biofilm assay showed that the biofilm biomasses of WT, Δ*crp*, and C*crp* were similar at 12 h; however, the Δ*crp* biofilm dispersed at 30 h, while the WT and C*crp* maintained a robust biofilm (Fig. 2a). The growth rates of the three strains showed that compared to WT and C*crp*, the growth rate of Δ*crp* was slightly slower at the early exponential phase (before 12 h), but Δ*crp* grew to a slightly higher cell density than the other strains after 12 h (Fig. 2b). The ratio of biofilm biomass to cell growth ($OD_{570}/OD_{600}$) of Δ*crp* was still significantly lower than that of the WT and C*crp* at 30 h (Supplementary Fig. 1b), suggesting that the statistically significant decline of the biofilm biomass of Δ*crp* was not caused by the

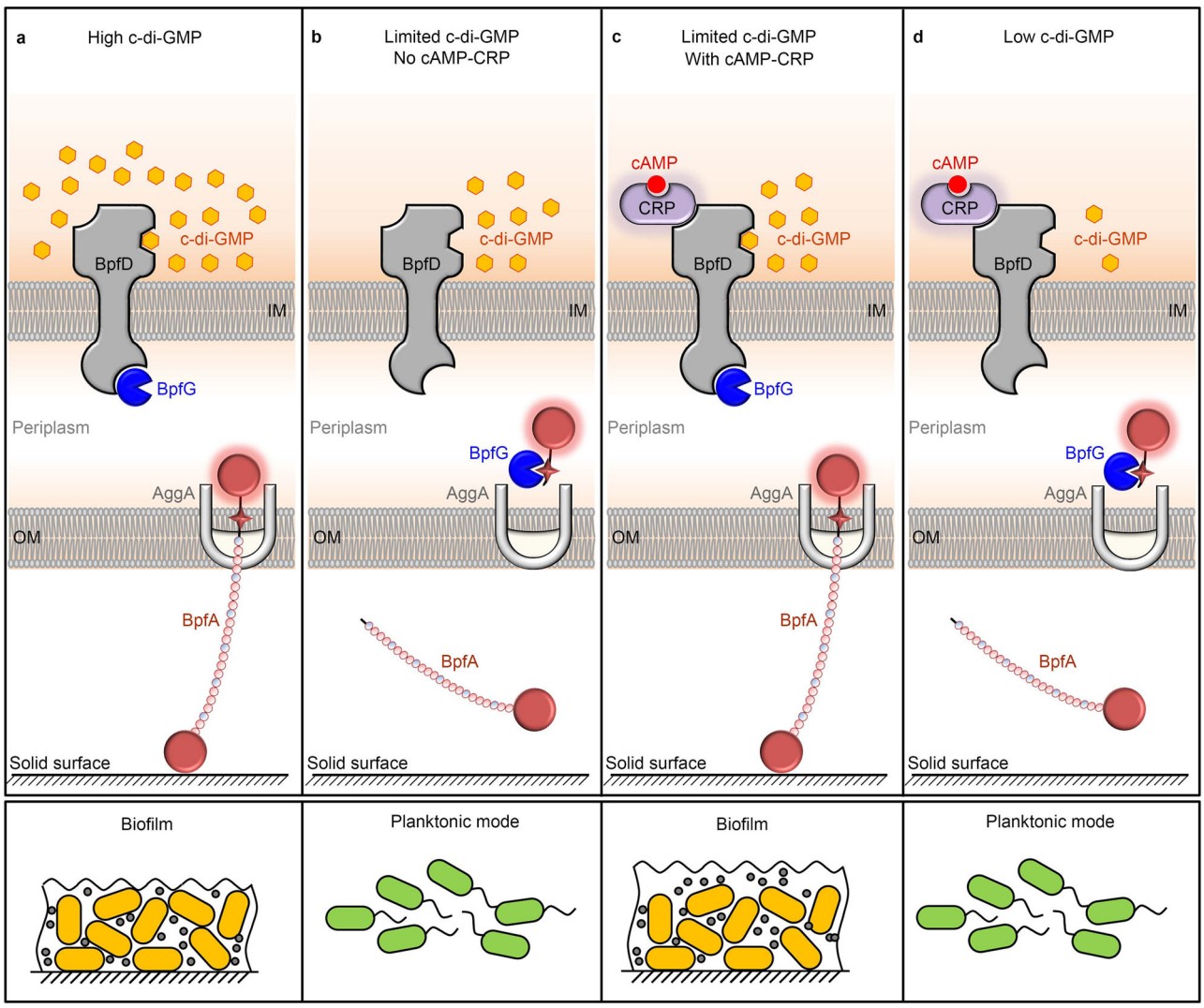

**Fig. 1 Pattern for BpfAGD system regulated by cAMP and c-di-GMP. a** A high intracellular c-di-GMP level activates the c-di-GMP effector BpfD to bind to and sequester the periplasmic protease BpfG, which prevents BpfA being processed and results in biofilm formation. **b** When intracellular c-di-GMP level is limited, BpfD cannot sequester BpfG, leaving BpfG free to process and release BpfA from the cell surface, leading to planktonic mode. **c** Although intracellular c-di-GMP level is limited, cAMP and c-di-GMP synergistically maintain the interaction between BpfD and BpfG through the direct interaction of their effector proteins, CRP and BpfD, thereby supporting biofilm maintenance. **d** When the intracellular c-di-GMP content is low and below a certain threshold concentration, the presence of cAMP-CRP cannot maintain the interaction between BpfD and BpfG, leading to biofilm dispersion. IM: Inner membrane, OM: Outer membrane.

change in cell growth. These results indicate that CRP is necessary to support the biofilm maintenance of *S. putrefaciens* CN32.

CRP is ubiquitous in bacteria and is widely known as a cAMP-dependent transcription factor. *S. putrefaciens* CN32 has three adenylate cyclases: CyaA (Sputcn32_3586), CyaB (Sputcn32_3104), and CyaC (Sputcn32_1140). In order to determine whether CRP regulates biofilm maintenance through forming a complex with cAMP, we created a mutant lacking all three adenylate cyclases ΔcyaAΔcyaBΔcyaC (named Δcya) and its complementation strain bearing a plasmid encoding all three adenylate cyclases CcyaA-CcyaB-CcyaC (named Ccya). The intracellular cAMP concentration was not detected in Δcya, and the intracellular cAMP concentration in Ccya was restored (Fig. 2c), which confirmed that Δcya is a cAMP-negative mutant. The biofilm assay showed that similar to Δcrp, the Δcya exhibited poor biofilm maintenance (Fig. 2d), while the deletion and complementation of three adenylate cyclase genes in Δcrp (ΔcrpΔcya and ΔcrpΔcyaCcya) did not influence the biofilm phenotype of Δcrp at 30 h (Fig. 2d). Meanwhile, the biofilm biomasses of WT, Δcya, Ccya, Δcrp, ΔcrpΔcya, and ΔcrpΔcyaCcya

were similar at 12 h (Supplementary Fig. 1c). These results indicate that cAMP-CRP acts as a complex to regulate biofilm maintenance. The cell growth of Δcya and Δcrp were similar (Fig. 2b), indicating that the biofilm differences between WT and Δcya were not caused by the change in cell growth. Thus, the biofilm maintenance of *S. putrefaciens* CN32 at 30 h was supported by the cAMP-CRP complex.

We deleted the cAMP PDE gene cpdA (ΔcpdA) to further confirm that CRP regulates biofilm maintenance dependent on cAMP. Although the deletion of cpdA increased the intracellular cAMP concentration (Fig. 2c), the significant decrease in the cell growth of ΔcpdA (Supplementary Fig. 1d) made it difficult to analyze the biofilm phenotype. Thus, to further investigate the influence of the increasing intracellular cAMP concentration on biofilm maintenance, 1 mM exogenous cAMP was added to the biofilm medium. The results showed that the addition of exogenous cAMP increased the intracellular cAMP concentration of all tested strains (Fig. 2e); however, the increase in the intracellular cAMP concentration only restored the biofilm

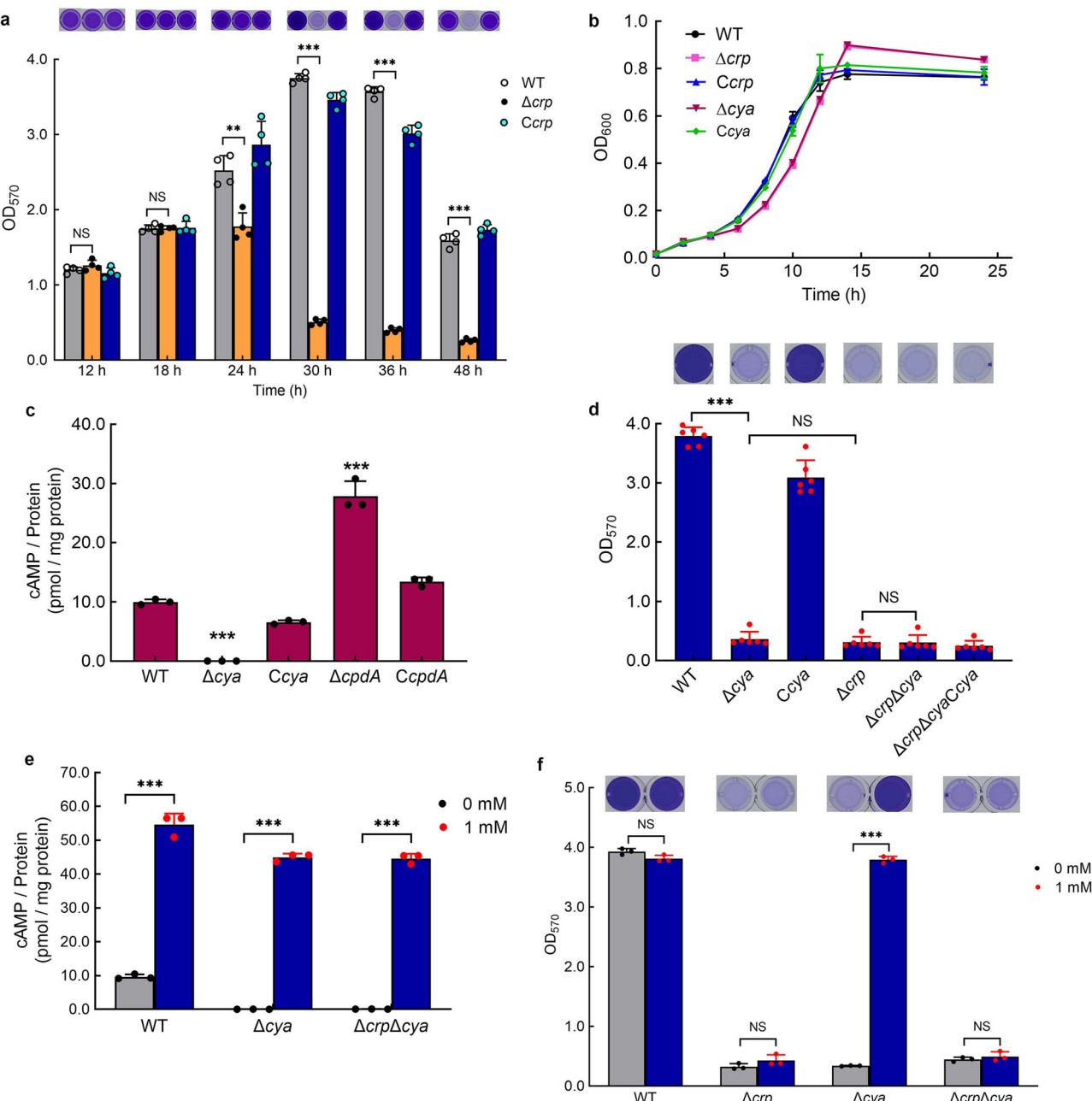

**Fig. 2 cAMP-CRP complex supports biofilm maintenance in *S. putrefaciens* CN32. a** Biofilm biomass ($n = 4$ independent samples). **b** Cell growth ($n = 3$ independent samples). **c** Intracellular cAMP concentration at 30 h ($n = 3$ independent samples). **d** Biofilm biomass at 30 h ($n = 6$ independent samples). **e** Intracellular cAMP concentration with the addition of 1 mM exogenous cAMP to the culture medium vs. control (no addition of exogenous cAMP [0 mM]) at 30 h ($n = 3$ independent samples). **f** Biofilm biomass with the addition of 1 mM exogenous cAMP to the culture medium vs. control at 30 h ($n = 3$ independent samples). Insets in (**a**, **d**, **f**) are the biofilm pictures of crystal violet dyeing. Data in (**a–f**) are shown as the mean ± SD. Two-sided Student's *t* test was used in (**a**, **c–f**) to analyze the statistical significance (NS: No significance. \*\*\**p* < 0.001). Source data are provided as a Source Data file.

maintenance of Δ*cya*, and exerted no effect on biofilm maintenance of Δ*crp* and Δ*crp*Δ*cya* at 30 h (Fig. 2f). These results further confirm that the biofilm maintenance is regulated by the cAMP-CRP complex.

**cAMP-CRP maintains biofilm independently of its regulation of *bpfA* transcription.** We next focused on investigating how cAMP-CRP regulates biofilm maintenance. It has been reported that large adhesive proteins play a critical role in biofilm formation or dispersion in several bacteria, including LapA in *P. fluorescens* and BpfA homolog in *Shewanella* spp.[25,42].

Previous studies have shown that *S. putrefaciens* CN32 exhibits defective biofilm formation due to the deletion of *bpfA*[51,52]. In this report, as Δ*cya*, Δ*crp* and Δ*crp*Δ*cya* exhibit similar biofilm biomasses to Δ*bpfA* at 30 h (Fig. 3a), we considered that whether *bpfA* is the target regulated by cAMP-CRP. Therefore, we deleted *bpfA* in these three mutants, and the result showed that the biofilm biomasses of Δ*cya*Δ*bpfA*, Δ*crp*Δ*bpfA* and Δ*crp*Δ*cya*Δ*bpfA* were similar to that of Δ*bpfA* (Fig. 3a). Furthermore, we found that although the addition of exogenous cAMP increased the intracellular cAMP concentration in all of the tested strains (Fig. 3b), the increase in intracellular cAMP concentration only restored the biofilm biomass of Δ*cya* to the WT level, but exerted

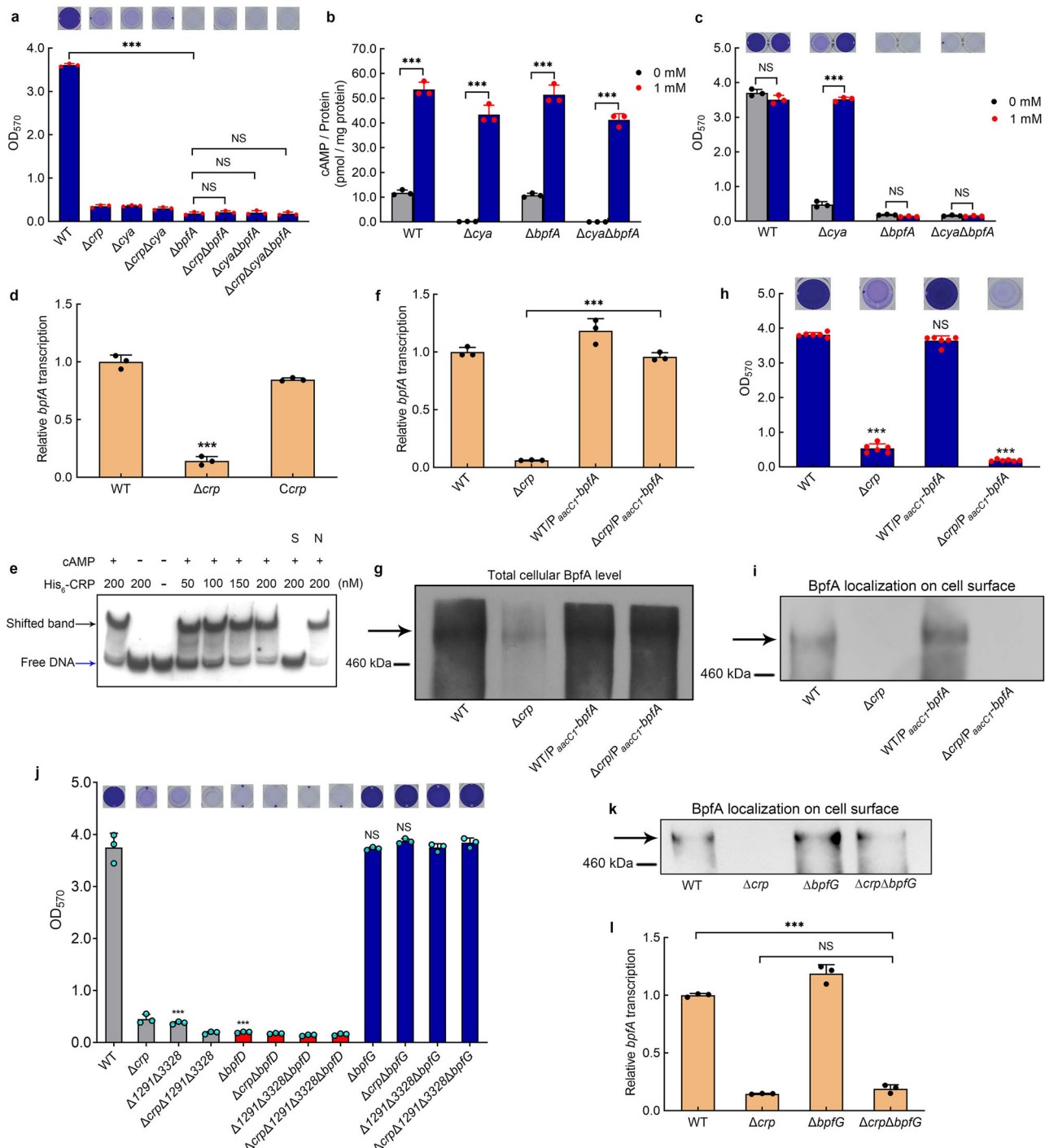

**Fig. 3 cAMP-CRP maintains biofilm independently of its regulation of *bpfA* transcription. a** Biofilm biomass ($n = 3$ independent samples). **b** Intracellular cAMP concentration with adding 1 mM exogenous cAMP to the culture medium vs. control ($n = 3$ independent samples). **c** Biofilm biomass with adding 1 mM exogenous cAMP to the culture medium vs. control ($n = 3$ independent samples). **d** Transcriptional analysis of *bpfA* ($n = 3$ independent samples). **e** EMSA of cAMP-CRP binding to *bpfA* promoter (*bpfA*-pro). The His$_6$-CRP (Lane–), labeled probe was incubated in the absence of His$_6$-CRP. The concentrations of His$_6$-CRP are shown above the figure. The binding specificity was confirmed by competitive assays with a 300-fold excess of unlabeled specific probe *bpfA*-pro (lane S) or unlabeled nonspecific competitor DNA (probe *recA*) (lane N). The cAMP (Lane –), labeled probe was incubated in the absence of cAMP. The cAMP (Lane +), labeled probe was incubated in 1 µM cAMP. **f** The comparison of *bpfA* transcription between the native *bpfA* promoter and the constitutive promoter P$_{aacC1}$ ($n = 3$ independent samples). **g** Western blotting detection of the total content of cellular BpfA protein. **h** Biofilm biomass ($n = 6$ independent samples). **i** Western blotting detection of BpfA localization on the cell surface. **j** Biofilm biomass ($n = 3$ independent samples). **k** Western blotting detection of BpfA localization on the cell surface. **l** Transcriptional analysis of *bpfA* ($n = 3$ independent samples). All strains used in (**a–d, f–l**) were cultured for 30 h in the biofilm state. Insets in (**a, c, h, j**) are the biofilm pictures of crystal violet dyeing. Data in (**a–d, f, h, j, l**) are shown as the mean ± SD. Two-sided Student's *t* test was used in (**a–d, f, h, j, l**) to analyze the statistical significance (NS: No significance. ***$p < 0.001$). Source data are provided as a Source Data file.

no effects on the biofilm biomass of $\Delta cya\Delta bpfA$ (Fig. 3c), indicating that the biofilm formation is poor as a result of the deletion of $bpfA$, irrespective of the change in intracellular cAMP concentration. These results suggest that BpfA is the downstream target of cAMP-CRP.

In bacteria, cAMP-CRP is known as a global transcription factor. Thus, we considered whether cAMP-CRP directly controls the transcription of $bpfA$. Compared to WT and C$crp$, the transcription of $bpfA$ in $\Delta crp$ was significantly down-regulated (Fig. 3d). The results of the electrophoretic mobility shift assay (EMSA) showed that in the presence of cAMP, CRP binds to the promoter region of $bpfA$ (Fig. 3e), indicating that cAMP-CRP directly triggers the transcription of $bpfA$. To further investigate whether cAMP-CRP supports biofilm maintenance through triggering the transcription of $bpfA$, the native promoter of the $bpfA$ operon was replaced by a constitutive promoter P$_{aacC1}$ in WT and $\Delta crp$. The result showed that the transcription of $bpfA$ in $\Delta crp$/P$_{aacC1}$-$bpfA$ was similar to that of WT and WT/P$_{aacC1}$-$bpfA$, and significantly higher than that in $\Delta crp$ (Fig. 3f), which is consistent with their total intracellular BpfA protein levels (Fig. 3g). If cAMP-CRP regulates biofilm maintenance dependent on promoting the transcription of $bpfA$, the biofilm biomass of $\Delta crp$/P$_{aacC1}$-$bpfA$ should be higher than that of $\Delta crp$ and close to that of WT. However, in practice, we found that increased transcription of $bpfA$ in $\Delta crp$/P$_{aacC1}$-$bpfA$ did not restore biofilm maintenance. The biofilm biomass of $\Delta crp$/P$_{aacC1}$-$bpfA$ was similar to that of $\Delta crp$ and was still significantly lower than that of WT and WT/P$_{aacC1}$-$bpfA$ at 30 h (Fig. 3h), which was consistent with the localization of BpfA on the cell surface (Fig. 3i, Supplementary Fig. 1e). Thus, cAMP-CRP regulates the biofilm maintenance independently of its regulation of $bpfA$ transcription.

We next investigate whether cAMP-CRP controls biofilm maintenance by retaining BpfA on the cell surface. As the retention of BpfA depends on the BpfAGD system (Fig. 1a), we deleted $bpfD$ and $bpfG$ in both WT and $\Delta crp$. Similar to $\Delta bpfA$, neither $\Delta bpfD$ nor $\Delta crp\Delta bpfD$ was able to form biofilm, while the biofilm biomasses of $\Delta bpfG$ and $\Delta crp\Delta bpfG$ at 30 h were similar to that of WT, and significantly higher than that of $\Delta crp$ (Fig. 3j). This finding suggests that the loss of BpfG restores the biofilm maintenance of $\Delta crp$ by preventing cleavage of BpfA from the cell surface. This was further confirmed by the immunoblot result showing that the localization of BpfA on the cell surface in $\Delta crp\Delta bpfG$ was restored to the WT level, which was significantly higher than that in $\Delta crp$ (Fig. 3k, Supplementary Fig. 1f). In addition, we found that the transcription of $bpfA$ in $\Delta crp\Delta bpfG$ is similar to that in $\Delta crp$ (Fig. 3l), demonstrating that the difference in BpfA localization on the cell surface of $\Delta crp$ and $\Delta crp\Delta bpfG$ is not caused by the loss of BpfG affecting $bpfA$ transcription. Taken together, these findings show that cAMP-CRP supports biofilm maintenance through retaining BpfA on the cell surface, rather than through regulating $bpfA$ transcription.

**cAMP-CRP maintains biofilm independently of its regulation of c-di-GMP level.** Previous studies have shown that the intracellular c-di-GMP level controls BpfA localization on the cell surface by regulating the BpfAGD system[55] (Fig. 1a). Therefore, we considered whether cAMP-CRP controls biofilm maintenance through regulating the intracellular c-di-GMP level. Our results showed that compared to WT, the intracellular c-di-GMP concentration in $\Delta crp$ and $\Delta cya$ decreased by ~25% (Fig. 4a). To test whether this 25% c-di-GMP decrease can cause biofilm dispersion of $\Delta crp$ and $\Delta cya$, we expressed DgcQ (formerly known as YedQ), a DGC of $E.$ $coli$ MG1655[63], in $\Delta crp$ and $\Delta cya$. The result showed that the introduction of DgcQ not only significantly increased

intracellular c-di-GMP concentration in both strains (Fig. 4a) but also largely restored biofilm maintenance (Fig. 4b). In addition, intracellular cAMP was still not detected in $\Delta cya$/p$dgcQ$ (Supplementary Fig. 1g), indicating that the biofilm maintenance restoration of $\Delta crp$/p$dgcQ$ and $\Delta cya$/p$dgcQ$ is due to the increase in intracellular c-di-GMP concentration, rather than the intracellular cAMP concentration. These results suggest that cAMP-CRP seems to modulate biofilm maintenance by regulating the transcription levels of some $dgc$/$pde$ genes.

To screen biofilm maintenance-related $dgc$/$pde$ genes whose transcriptions are regulated by cAMP-CRP, the transcription levels of all 47 $dgc$/$pde$ genes in the genome of $S.$ $putrefaciens$ CN32 were compared in WT and $\Delta crp$ at 30 h (Fig. 4c, d, Supplementary Table 1). The results showed that cAMP-CRP regulates the transcription of 41 of the $dgc$/$pde$ genes. To increase screening efficiency, we selected 11 genes whose transcription level in $\Delta crp$ changed by more than 2.5-fold compared to WT, and only the biofilm biomasses of $\Delta 1291$ and $\Delta 3328$ were slightly lower than that of the WT at 30 h (Fig. 4e, f, Supplementary Fig. 2). The biofilm of the double mutant $\Delta 1291\Delta 3328$ was poor, while the WT and C$1291$C$3328$ maintained robust biofilms at 30 h (Fig. 4g). And the intracellular c-di-GMP concentration in $\Delta 1291\Delta 3328$ was significantly lower than that in the WT and C$1291$C$3328$ (Fig. 4h). Moreover, the intracellular c-di-GMP concentration and the biofilm biomass of a site-directed mutant C$1291$(GGAAF)C$3328$(GGAAF) were similar to those of $\Delta 1291\Delta 3328$ at 30 h (Fig. 4g, h). These results indicate that Sputcn32_1291 and Sputcn32_3328 have DGC activity in vivo and participate in supporting biofilm maintenance. Besides, the results of biochemical assays showed that Sputcn32_1291 could catalyze GTP to produce c-di-GMP in vitro (Fig. 4i). However, we failed to detect the DGC or PDE activity of Sputcn32_3328 in vitro, which may be due to Sputcn32_3328 belongs to a three-component regulatory system, DGC activity of Sputcn32_3328 may depend on its cognate proteins in vivo. We confirmed that both Sputcn32_1291 and Sputcn32_3328 act as DGCs in vivo to regulate biofilm maintenance; this is not only due to the presence of the conserved GGDEF domain of DGC activity in both proteins, which influence the intracellular c-di-GMP concentration, but also due to the fact that the heterologous expression of $dgcQ$ in $\Delta 1291\Delta 3328$ restored the biofilm biomass to the WT level (Fig. 5a) by increasing the intracellular c-di-GMP concentration (Fig. 5b).

As cAMP-CRP regulates the transcription of Sputcn32_3328 and Sputcn32_1291 genes (Fig. 4c, d) and Sputcn32_1291 and Sputcn32_3328 act as DGCs to support biofilm maintenance (Fig. 4g, h), it seems that cAMP-CRP regulates biofilm maintenance through modulating the intracellular c-di-GMP level. However, Fig. 4g, h show some contradictory results. First, the intracellular c-di-GMP concentration of $\Delta crp$ was significantly higher than that of $\Delta 1291\Delta 3328$ (Fig. 4h); however, both $\Delta crp$ and $\Delta 1291\Delta 3328$ showed similarly poor biofilm maintenance, which was significantly lower than that of the WT (Fig. 4g). Second, if cAMP-CRP regulated biofilm maintenance by modulating the intracellular c-di-GMP level, the increase in intracellular c-di-GMP concentration of $\Delta crp$ to the WT level should restore its biofilm biomass to the WT level. However, in practice, we found that the intracellular c-di-GMP concentration in $\Delta crp\Delta 1291\Delta 3328$C$1291$C$3328$ was significantly higher than that in $\Delta crp$, and even higher than that of the WT (Fig. 4h), but its biofilm biomass was slightly higher than that of $\Delta crp$, and still significantly lower than that of WT (Fig. 4g). Thus, the biofilm maintenance supported by cAMP-CRP may not be totally dependent on regulating the intracellular c-di-GMP level. Besides, although the transcriptional levels of Sputcn32_1291 and Sputcn32_3328 genes in $\Delta crp$ were significantly lower than those

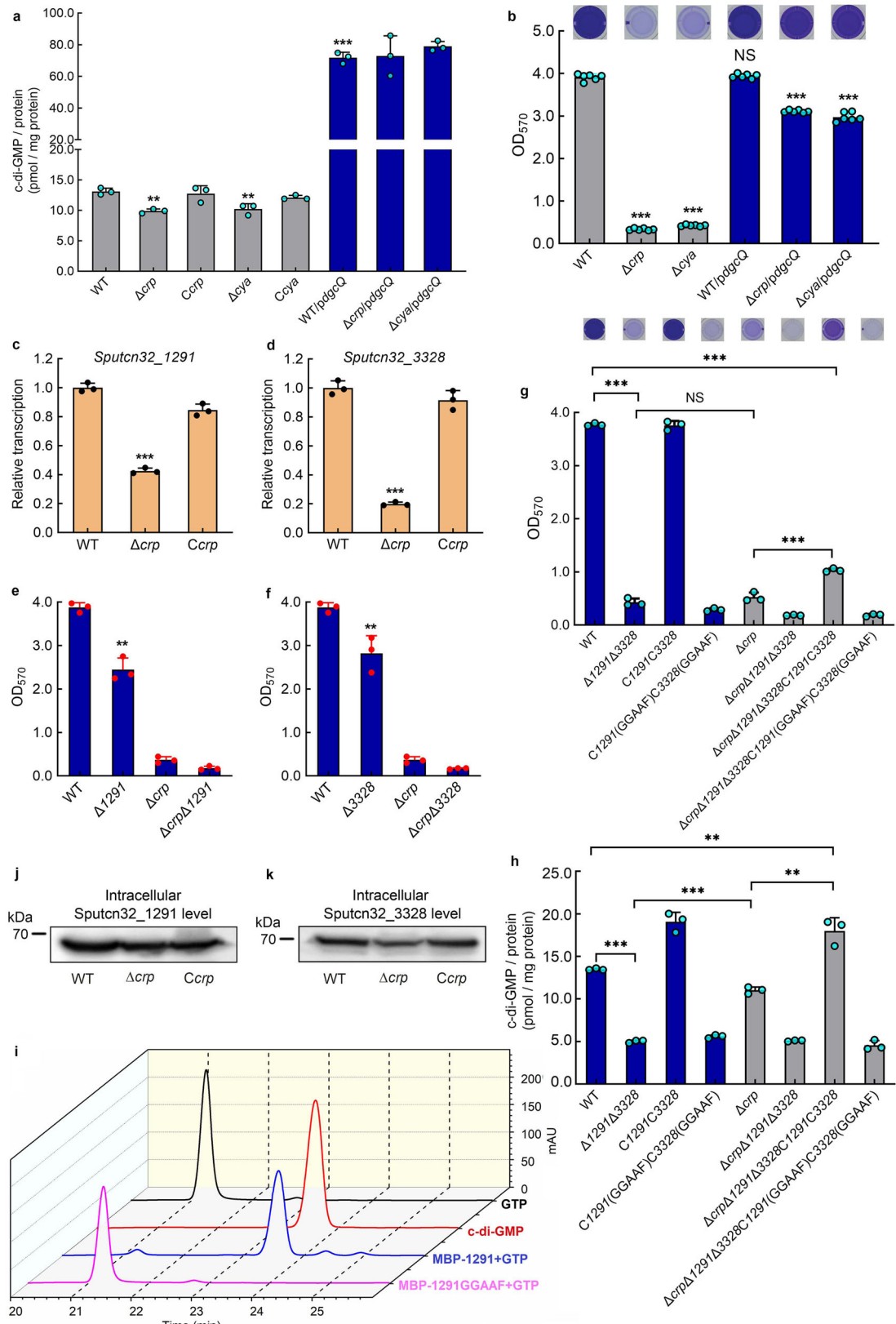

in WT at 30 h (Fig. 4c, d), the intracellular level of both proteins at 30 h showed no significant difference or only changed slightly between WT and Δ*crp* (Fig. 4j, k). Thus, cAMP-CRP regulates biofilm maintenance independently of its regulation of the transcription of both *dgc* genes. The finding that the expression of DgcQ in Δ*crp* can partly restore its biofilm maintenance

(Fig. 4b) is probably due to DgcQ significantly enhancing the intracellular c-di-GMP concentration (Fig. 4a), which acts as a remedial mechanism in the absence of cAMP-CRP.

To further confirm this hypothesis, we significantly increased the intracellular c-di-GMP concentration of Δ*1291Δ3328* and Δ*crpΔ1291Δ3328* through over-expressing *dgcQ* in both mutants

**Fig. 4 cAMP-CRP maintains biofilm independently of its regulation of c-di-GMP level. a** Intracellular c-di-GMP concentration ($n = 3$ independent samples). **b** Biofilm biomass ($n = 6$ independent samples). **c, d** Transcriptional analysis of *Sputcn32_1291* and *Sputcn32_3328* ($n = 3$ independent samples). **e, f, g** Biofilm biomass ($n = 3$ independent samples). **h** Intracellular c-di-GMP concentration ($n = 3$ independent samples). All strains used in (**a–h**) were cultured for 30 h in the biofilm state. **i** DGC enzymatic assays showing the DGC activity of Sputcn32_1291, GTP and c-di-GMP were used as standard samples in HPLC. **j, k** Western blotting detection of the total protein content of cellular Sputcn32_1291 and Sputcn32_3328 at 30 h. Insets in **b, g** are the biofilm pictures of crystal violet dyeing. Data in (**a–h**) are presented as the mean ± SD. Two-sided Student's *t* test was used in (**a–h**) to analyze the statistical significance (NS: No significance. **$p < 0.01$. ***$p < 0.001$). Source data are provided as a Source Data file.

(Fig. 5b). The biofilm assay showed that only the biofilm biomass of Δ1291Δ3328/p*dgcQ* was restored to the WT level, but the biofilm biomass of Δ*crp*Δ1291Δ3328/p*dgcQ* was still lower than that of WT (Fig. 5a), indicating that the deletion of *crp* significantly weakened the restoration effect of increased intracellular c-di-GMP level on biofilm maintenance. Taken together, although increasing the intracellular c-di-GMP level can partly restore the decrease in biofilm maintenance caused by the absence of cAMP-CRP, cAMP-CRP regulates biofilm maintenance independently of its regulation of the intracellular c-di-GMP concentration.

**cAMP-CRP and c-di-GMP maintain biofilm by retaining BpfA on the cell surface.** To further uncover the mechanism by which cAMP-CRP and c-di-GMP regulate biofilm maintenance, we deleted *bpfA*, *bpfD* and *bpfG* in both Δ1291Δ3328 and Δ*crp*Δ1291Δ3328. The results showed that similar to the deletion of *bpfA* or *bpfD* in Δ*crp*, deletion of *bpfA* or *bpfD* blocked the biofilm formation of Δ1291Δ3328 and Δ*crp*Δ1291Δ3328 (Figs. 3j, 5a), which is consistent with the regulatory pattern of the BpfAGD system as shown in Fig. 1a. Moreover, the deletion of *bpfG* restored the biofilm maintenance of Δ*crp*, Δ1291Δ3328 and Δ*crp*Δ1291Δ3328 to the WT level (Fig. 3j), suggesting that both c-di-GMP and cAMP-CRP regulate biofilm maintenance through controlling the BpfAGD system.

To further confirm this finding, the influence of cAMP-CRP and c-di-GMP on the localization of BpfA on the cell surface was analyzed. We first replaced the native promoter region of the *bpfA* operon with the constitutive promoter P*aacC1* in WT, Δ*crp*, Δ1291Δ3328, Δ*crp*Δ1291Δ3328, O1291O3328, Δ*crp*O1291O3328, Δ1291Δ3328/p*dgcQ*, and Δ*crp*Δ1291Δ3328/p*dgcQ* to eliminate the differences in *bpfA* transcription caused by changes in the c-di-GMP and cAMP-CRP levels. The qRT-PCR results showed that the promoter replacement of the *bpfA* operon reduced the transcription differences in the *bpfA* operon among these eight strains (Supplementary Fig. 3a), but that the replacement did not change the biofilm maintenance phenotype of the corresponding strains (Supplementary Fig. 3b). We next examined the BpfA localization on the cell surface of strains with P*aacC1*-*bpfA* replacement using immunoblot. The result showed that BpfA was almost undetectable on the cell surface of Δ*crp*, Δ1291Δ3328, and Δ*crp*Δ1291Δ3328 (Fig. 5c, Supplementary Fig. 3c), indicating that the absence of cAMP-CRP and the reduction in intracellular c-di-GMP concentration causes the release of BpfA from the cell surface. Besides, the increase in the intracellular c-di-GMP concentration retains the BpfA localization on the cell surface (see the mutants O1291O3328, Δ*crp*O1291O3328, Δ1291Δ3328/p*dgcQ* and Δ*crp*Δ1291Δ3328/p*dgcQ* in Fig. 5c, Supplementary Fig. 3c), thereby supporting biofilm maintenance (Fig. 5a). Thus, both cAMP-CRP and c-di-GMP support biofilm maintenance by retaining BpfA on the cell surface.

**cAMP-CRP and BpfD interaction retains BpfA on the cell surface.** Finally, we sought to investigate how cAMP-CRP retains BpfA on the cell surface. As shown in Fig. 1a, BpfA localization on the cell surface is regulated by the interaction between BpfD

(c-di-GMP effector) and BpfG[55]. Thus, the interaction between BpfD and BpfG in WT, Δ*crp*, Δ1291Δ3328, and Δ*crp*O1291O3328 was detected by co-immunoprecipitation (Co-IP). In all four strains, the native promoter region of the *bpfA* operon was replaced with the constitutive promoter P*aacC1*, a chromosomal C-terminal 3×Flag-tag was attached to BpfD, and a chromosomal near C-terminal 1×HA-tag was attached to BpfG, all of which did not influence the biofilm maintenance compared to that of their original strains (Supplementary Fig. 3d). After the replacement of *bpfA* operon promoter, the protein levels of BpfD and BpfG were similar in the four strains (input bands of Fig. 5d), indicating that the differences in BpfA localization on the cell surface among these strains were not caused by the change in total protein of BpfD and BpfG. The Co-IP result showed that the interaction between BpfD and BpfG was greatly weakened in Δ*crp* and Δ1291Δ3328 compared to that in WT and Δ*crp*O1291O3328, indicating that both cAMP-CRP and c-di-GMP regulate the interaction between BpfD and BpfG (Fig. 5d, Supplementary Fig. 3e). This finding is consistent with the previously reported conclusion that c-di-GMP is essential in promoting the interaction between BpfD and BpfG[55,64].

We next consider how cAMP-CRP regulates the interaction between BpfD and BpfG. It has been reported that GcbC, a DGC, enhances the interaction between LapD and LapG by directly binding to LapD in *P. fluorescens*[65]. This led us to consider that there may be a CRP-regulated protein or CRP-self, which acts as the enhancer to promote the interaction between BpfD and BpfG. The results of Co-IP, GST-pull down, and microscale thermophoresis (MST) experiments all showed that CRP directly interacted with BpfD, and that their interaction does not require the presence of cAMP (Fig. 5e–g, Supplementary Fig. 3f). However, as the above results indicate that cAMP-CRP acts as a complex to regulate biofilm maintenance (Fig. 2d, f), we considered whether cAMP is necessary to enhance the interaction between BpfD and BpfG. A site-directed mutant protein CRP-R84L (Arg-84→Leu), which has lost the ability to bind cAMP[3] (Supplementary Fig. 3g), still interacted with BpfD (Figs. 5e, 6a). However, the interaction between BpfD and BpfG, BpfA localization on the cell surface, and the biofilm maintenance of C*crp*-R84L were similar to those in Δ*crp* (Fig. 6b–d, Supplementary Fig. 3h). These results indicate that although CRP physically interacts with BpfD (Fig. 5e), the ability of CRP to enhance the interaction between BpfD and BpfG is lost in the absence of cAMP (Fig. 6c, Supplementary Fig. 4a–c).

**Pattern for cAMP and c-di-GMP synergistically maintaining biofilm.** Based on the above results, we propose a pattern to explain the phenotype of the different mutants. As shown in Fig. 1c, cAMP and c-di-GMP synergistically maintain the interaction between BpfD and BpfG through the direct interaction of their effector proteins, CRP and BpfD, thereby supporting biofilm maintenance. The presence (such as WT) or absence (such as Δ*crp*) of the cAMP-CRP complex determines the biofilm maintenance (Fig. 1c) or biofilm dispersion (Fig. 1b), respectively. Significant increase in the intracellular c-di-GMP concentration in Δ*crp* (such as Δ*crp*O1291O3328, Fig. 1a) maintains the interaction

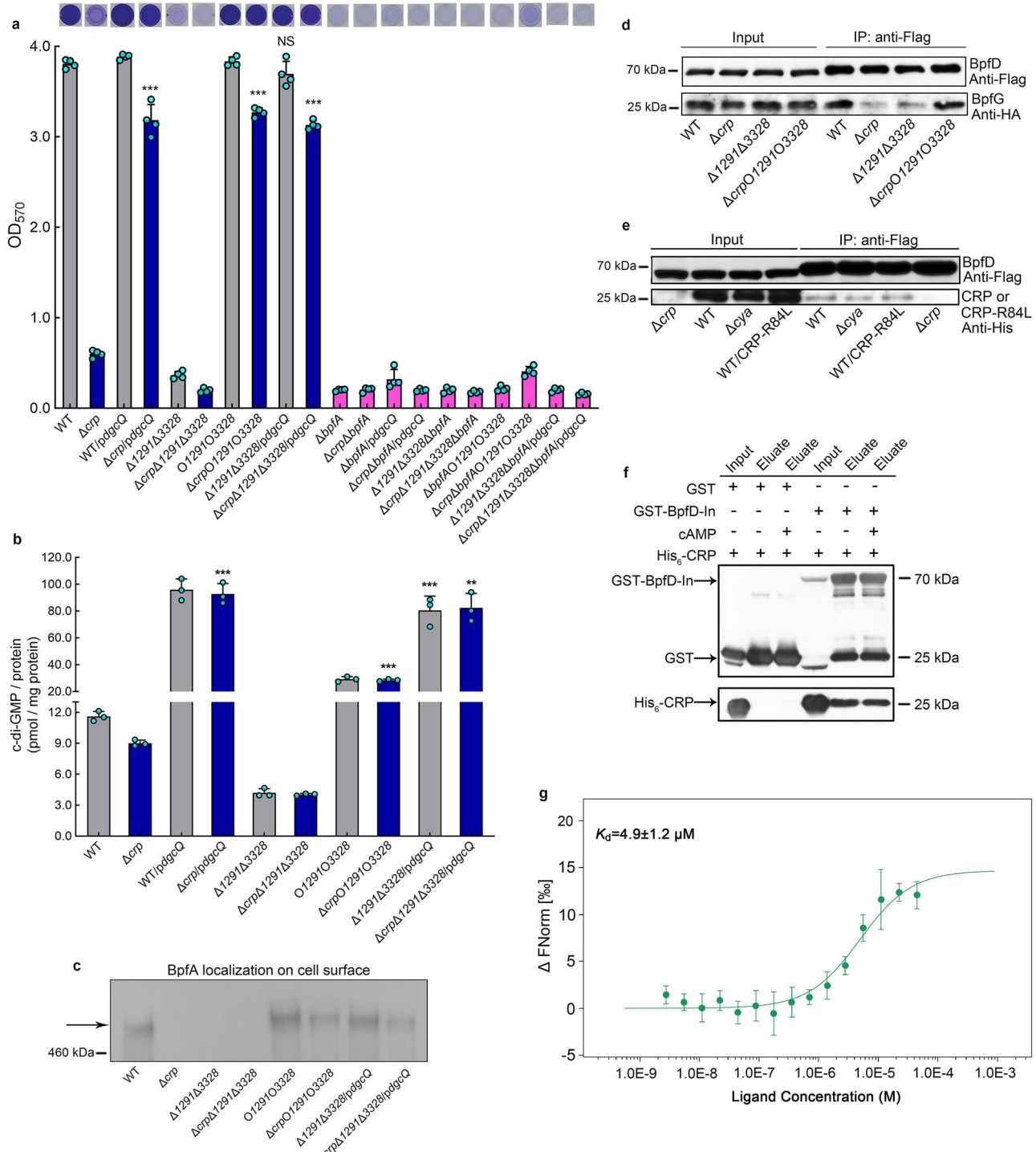

**Fig. 5 Direct interaction between cAMP-CRP and BpfD retains BpfA localization on the cell surface. a** Biofilm biomass ($n = 3$ independent samples). **b** Intracellular c-di-GMP concentration ($n = 3$ independent samples). **c** Western blotting detection of BpfA localization on the cell surface. **d** Co-IP to analyze the interaction between BpfD and BpfG in vivo. **e** Co-IP to analyze the interaction between BpfD and CRP/CRP-R84L in vivo. All strains used in (**a–e**) were cultured for 30 h in the biofilm state. **f** GST pull-down assay showing the interaction between BpfD intracellular domains and CRP in vitro. **g** MST showing the interaction between BpfD intracellular domains and CRP in vitro, data are shown as the mean ± SD ($n = 3$ independent samples). The native promoter region of the *bpfA* operon of the strains used in (**c–e**) was replaced by the constitutive promoter P_{aacC1}. Insets in (**a**) are the biofilm pictures of crystal violet dyeing. Data in (**a**, **b**) are shown as the mean ± SD. Two-sided Student's *t* test was used in (**a**, **b**) to analyze the statistical significance (NS: No significance. **$p < 0.01$. ***$p < 0.001$). Source data are provided as a Source Data file.

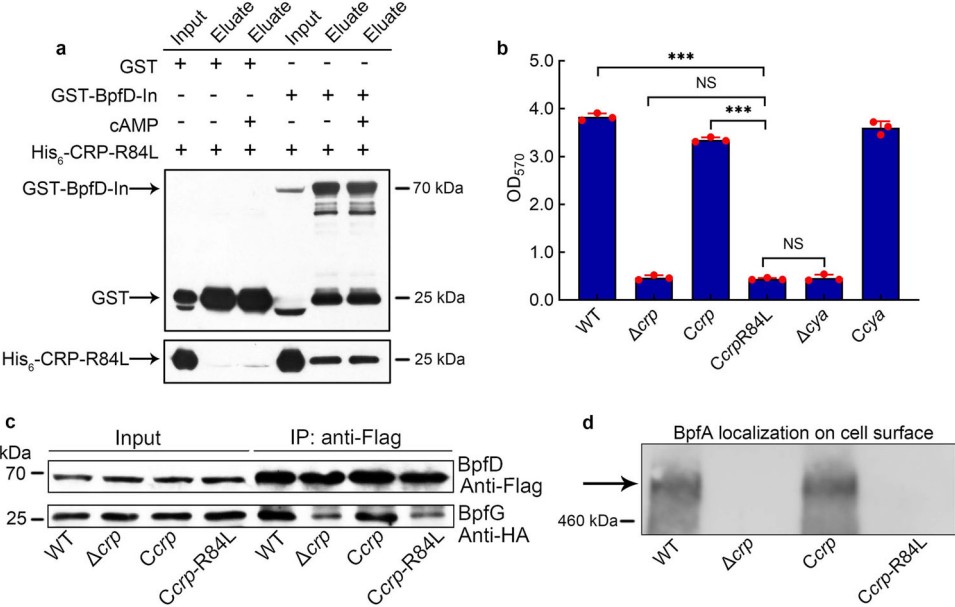

**Fig. 6 cAMP is not necessary for physical interaction between CRP and BpfD, but it is essential to help CRP enhance the interaction between BpfD and BpfG. a** GST pull-down assay showing the interaction between BpfD intracellular domains and CRP-R84L in vitro. **b** Biofilm biomass ($n = 3$ independent samples). **c** Co-IP to analyze the interaction between BpfD and BpfG in vivo. **d** Western blotting detection of BpfA localization on the cell surface. All strains used in (**a, c, d**) were cultured for 30 h in the biofilm state. The native promoter region of the *bpfA* operon of the strains used in (**c, d**) was replaced by the constitutive promoter $P_{aacC1}$. Data in (**a**) are shown as the mean ± SD. Two-sided Student's *t* test was used in (**a**) to analyze the statistical significance (NS: No significance. ***$p < 0.001$). Source data are provided as a Source Data file.

between BpfD and BpfG (Fig. 5d), which retains BpfA on the cell surface (Fig. 5c, Supplementary Fig. 3c), thereby greatly remedying the biofilm maintenance of Δ*crp* (Fig. 5a). When the intracellular c-di-GMP content is low and below a certain threshold concentration (such as Δ*1291*Δ*3328*, Fig. 1d), the interaction between BpfD and BpfG cannot be maintained (Fig. 5d). As a result, BpfA is released from the cell surface (Fig. 5c, Supplementary Fig. 3c), leading to biofilm dispersion (Figs. 1d, 5a). Moreover, significant increase in the intracellular c-di-GMP concentration (Fig. 4a) exerts no effect on biofilm maintenance of WT (Fig. 4b); this is probably due to the fact that, in WT, as BpfG is already inhibited enough by BpfD in the presence of cAMP-CRP, further increasing the intracellular c-di-GMP level has no additive effect. The above conclusions are further confirmed by Supplementary Figs. 1g, 4d–f.

The relationships between intracellular c-di-GMP level and bacterial lifestyles have been well-established. Specially, a high intracellular c-di-GMP level is associated with biofilm formation, while a low intracellular c-di-GMP level tends to facilitate a planktonic lifestyle. However, several reports demonstrate that some regulators act as additional tool for fine-tuning such an important cellular molecular mechanism by cross-talking with c-di-GMP[66]. Our results reveal that in limited intracellular c-di-GMP level condition, cAMP-CRP plays a determinant function in the regulation of biofilm maintenance in *S. putrefaciens* CN32 (Fig. 1b, c), which underlines the complexity of bacterial second messenger regulation again. In summary, in *S. putrefaciens* CN32, cAMP-CRP acts as not only global transcription factor to regulate physiological metabolism but also post-translation regulator to participate in biofilm maintenance by cross-regulating with second messenger c-di-GMP.

## Discussion

Bacteria have two growth modes: biofilm, which is considered as a dominant bacterial growth mode in nature, and planktonic growth[26,67]. In the biofilm developmental process, initial

attachment and dispersion are widely referred to as promising avenues for biofilm control because both are key steps in the transition between motile planktonic and sessile biofilm lifestyles[22,23]. Although there has been an abundance of studies on the regulation of initial attachment in multiple bacteria, less is known about the maintenance of mature biofilm and dispersion[29,67]. Recently, two PDEs, RmcA and MorA, have been shown to regulate biofilm maintenance of *P. aeruginosa*, indicating that c-di-GMP plays a significant role in regulating biofilm maintenance[24]. Compared to c-di-GMP, the function of cAMP in biofilm formation seems more ancillary in most bacteria[62]. However, in this report, cAMP and c-di-GMP were found to play an equally important role and synergistically regulate biofilm maintenance in *S. putrefaciens* CN32, suggesting that the importance of cAMP in regulating biofilm development in bacteria may be underestimated in previous studies.

Several bacteria possess large adhesins localized on their cell surface, which assist with adherence to biotic or abiotic surfaces. Such adhesins are commonly controlled by a Lap system whose underlying mechanism has been well-established in *P. fluorescens*[42]. In *S. putrefaciens* CN32, a BpfAGD system that belongs to the Lap system controls biofilm formation by responding to intracellular c-di-GMP levels[55]. As shown in Fig. 1a, a high intracellular c-di-GMP level increases the interaction between BpfD and BpfG, which retains BpfA on the cell surface, thereby promoting initial biofilm formation[51,52,55]. Thus, we originally speculated that the cAMP-CRP complex may control biofilm maintenance by regulating the transcription of the *bpfA* operon and/or *dgc/pde* genes given that cAMP-CRP is widely recognized as a transcription factor. Indeed, the transcription of the *bpfA* operon and two key *dgc* genes in Δ*crp* were significantly lower than those in WT. However, further investigations showed that the reduction in the transcription of *bpfA* operon and *dgc* genes does not play a decisive role in the poor biofilm maintenance of Δ*crp*.

Further study showed that cAMP-CRP and c-di-GMP regulate BpfA localization on the cell surface. We next sought to determine how cAMP-CRP supports biofilm maintenance through regulating BpfAGD system. In *P. fluorescens*, the Lap system not only responds to the total intracellular c-di-GMP level, but is also specifically controlled by local c-di-GMP signaling[42]. GcbC is a DGC localized on the inner membrane, which contributes to the local c-di-GMP pool and is not responsible for the total intracellular c-di-GMP level. Citrate increases the c-di-GMP synthesis capability of GcbC and stimulates the physical interaction between GcbC and LapD[65]. Subsequently, GcbC synthesizes c-di-GMP and physically delivers c-di-GMP to LapD by contacting with LapD[42,65]. Thus, the Lap system responds to citrate signals and promotes biofilm formation in a GcbC-dependent manner[42]. The enhancement of the interaction between LapD and LapG by GcbC suggests that in *S. putrefaciens* CN32, there may be a CRP-regulated protein or CRP-self, which greatly enhances the interaction between BpfD and BpfG by directly contacting with BpfD. Surprisingly, through Co-IP, GST-pull down, and MST experiments, we found that CRP directly interacts with the c-di-GMP effector BpfD (Fig. 5e–g) and enhances the interaction between BpfD and BpfG in the presence of cAMP (Fig. 6c), thereby supporting biofilm maintenance. We describe a regulatory pattern that cAMP-CRP acts as a post-translation regulator to modulate biofilm maintenance.

When cAMP-CRP acts as a transcription factor, the amino acid residue of CRP binding to cAMP is Arg-83 in *E. coli*[3], which is Arg-84 in *S. putrefaciens* CN32 (Supplementary Fig. 3g). In this report, we found that although the CRP-R84L protein still interacted with BpfD (Figs. 5e, 6a), its function in regulating biofilm maintenance was lost (Fig. 6b–d). This not only indicates that CRP and cAMP must form a complex to perform regulatory function but also suggests that cAMP-CRP complex functions in the same conformation irrespective of acting as a transcription factor or a post-translation regulator. Extensive experiments are required to prove this hypothesis in future studies. Besides, heterologous complementation CRP (88% sequence identity with CN32 CRP) of *E. coli* MG1655 in Δ*crp* can partly restore the biofilm maintenance, but it cannot restore the biofilm biomass in Δ*crp*Δ*cya* (Supplementary Fig. 4g), indicating that the homologous CRP in *E. coli* has a function similar to CN32 CRP. Taken together, the above findings may suggest that the regulatory pattern of CRP as post-translation regulator is ubiquitous in bacteria. Future studies should pay more attention to the ability of cAMP-CRP to act as a post-translation regulator in bacteria.

As the biofilm biomasses of WT and Δ*crp* (Δ*cya*) were similar at 12 h (Fig. 2a, Supplementary Fig. 1c), it seems that cAMP-CRP did not participate in the early biofilm developmental stage. Similar to the Lap system of *P. fluorescens*, the BpfAGD system not only regulates initial attachment[55] but also controls biofilm dispersion of *S. putrefaciens* CN32. If cAMP-CRP can participate in biofilm maintenance through regulating the BpfAGD system, why is cAMP-CRP not involved in the early biofilm developmental stage? In *P. fluorescens* Pf0-1, similar to GcbC, another DGC, GcbB, is also involved in controlling biofilm formation through interacting with LapD, indicating multiple proteins regulate biofilm formation depending on interaction with LapD[42]. Thus, we speculated that although the BpfAGD system is involved in the entire biofilm developmental process, its upstream regulator varies depending on the different biofilm developmental stages. CRP interacts with BpfD in the later stage, and there may be another GcbC-like protein that participates in the regulation of initial attachment. In future, extensive experiments are required to explore more upstream regulators of the BpfAGD system at different biofilm developmental stages.

In bacteria, multiple nucleotide second messengers, such as cAMP, c-di-GMP, and guanosine tetra- and pentaphosphate ((p)ppGpp), have been reported to participate in the modulation of fundamental physiological processes[28]. Moreover, various second messengers coordinate multiple physiological metabolisms by cross-regulating each other[68,69]. For instance, cAMP has been reported to regulate the transcription of some genes encoding c-di-GMP receptors or DGCs/PDEs[59–61]. Furthermore, c-di-GMP and (p)ppGpp competitively bind to a common effector to control *Caulobacter crescentus* transition between swarmer and stalked lifestyles[70]. In this study, we describe a regulatory pattern whereby cAMP and c-di-GMP synergistically regulate biofilm maintenance by interaction of their effectors, which enriches the cross-regulation patterns between multiple second messengers. These findings are of great significance for understanding how bacteria intersect and integrate signals of second messengers. In addition, cAMP is ubiquitous in animals, plants, and microbes, and is known as a universal second messenger of organisms. It has been reported that pathogenic bacteria have evolved strategies to manipulate host cAMP concentrations[71]. In turn, cAMP not only is synthesized by bacteria themselves but also can enter the bacterial cells from their live environments[72]. Therefore, this study will assist with understanding the interaction between bacteria biofilms and host organisms.

## Methods

**Bacterial strains and growth conditions.** *E. coli* strains were grown in Luria-Bertani broth (LB) at 37 °C. *S. putrefaciens* CN32 WT and its derivative strains were cultured at 30 °C in LB broth or modified M1 defined minimal medium (MM1) containing 20 mM sodium lactate, 30 mM HEPES, 1.34 mM KCl, 28.04 mM NH$_4$Cl, 4.35 mM NaH$_2$PO$_4$, 7.5 mM NaOH, and 0.68 mM CaCl$_2$ supplemented with trace amounts of amino acids, minerals, and vitamins[51]. The cell growth of CN32 and its derivatives was tested in MM1 medium without addition 0.68 mM CaCl$_2$. When necessary, 50 µg ml⁻¹ kanamycin (Km) was supplemented in medium for CN32 derivatives. The strains and plasmids used in this study are listed in Supplementary Table 2, and the primers are listed in Supplementary Table 3. The kits used for the isolation and purification of DNA were purchased from Tiangen Biotech (China). The enzymes for molecular manipulation were purchased from New England BioLabs (USA) and Thermo Fisher Scientific (USA).

**Transposon mutagenesis and mapping the transposon insertion.** Transposon mutagenesis was performed by biparental conjugation between *S. putrefaciens* CN32 receptor and *E. coli* UQ3022 donor (harboring a plasmid pRL27)[51,73]. *S. putrefaciens* CN32 and *E. coli* UQ3022 were cultured in LB medium for 12 h. The cells were collected by centrifugation at 10,000 *g* for 30 s and washed twice using LB medium. Then, 50 µl mixtures of two strains were spot inoculated on LB agar plate. After incubation at 30 °C for 6 h, the cells were resuspended using 1 ml LB medium. A transposon insertion library was obtained by plating 100 µl of cell suspension on LB agar plate containing 20 µg ml⁻¹ tellurite and 50 µg ml⁻¹ kanamycin. Then, the plates were incubated at 30 °C for 36 h, and the black colonies were selected for biofilm formation assay. To identify the location of transposon insertion, the chromosomal DNA was extracted and digested with *Bam*HI or *Spe*I. The resulting fragments were self-ligated and transformed into *E. coli* UQ3021 cells. Subsequently, the transposon junction plasmids were extracted from the selected transformants and sequenced using primers Tn5-seq1/Tn5-seq2 (Supplementary Table 3) to reveal the location of transposon insertion.

**Construction of deletion mutants and complementation strains.** In-frame deletion mutants were generated using an established method[51]. To construct the *crp* deletion mutant, a 1164-bp upstream fragment (−1032 to +132 bp relative to the *crp* start codon) and a 1174 bp downstream fragment (+625 to +1159 bp relative to the *crp* start codon) were amplified using (primer pairs *crp*-5F/*crp*-5R and *crp*-3F/*crp*-3R, respectively). The homologous arms were digested by *Eco*RI/*Xba*I and *Xba*I/*Hin*dIII, and both were cloned into the *Eco*RI/*Hin*dIII-digested vector pK19*mobsacB*[74] to yield pK19*crp*UD. The pK19*crp*UD was introduced into *S. putrefaciens* CN32 by conjugation with the help of pRK2013[75]. The Δ*crp* deletion mutant was verified by PCR, using the following primer pairs *crp*-UF/*crp*-DR, *crp*-OF/*crp*-DR, *crp*-UF/*crp*-OR, and *crp*-INF/*crp*-INR. A similar strategy was applied for generating Δ*cya*, Δ*cpdA*, Δ*bpfA*, Δ*bpfD*, Δ*bpfG*, Δ*0133*, Δ*0654*, Δ*1291*, Δ*1365*, Δ*1412*, Δ*1858*, Δ*1934*, Δ*3319*, Δ*3328*, and Δ*3598* deletion mutants. The complementation strains were generated using plasmid pBBR1MCS-2[51,76]. All of the resulting mutants and complementation strains were verified by PCR and DNA sequencing.

**Construction of tagged transformant and replacement of P$_{bpfA}$.** The nucleotide sequences encoding tags were inserted into the corresponding location of the target gene in the genome. To construct a C-terminal 3×Flag-tagged BpfD transformant,

the 3′-terminus region of the *bpfD* gene, including its upstream and downstream flanks was amplified by the primers BpfD-C5F/BpfD-C3R and cloned into pK19*mobsacB* to yield an intermediate plasmid pK19-BpfD-Cter. The pK19-BpfD-Cter was linearized by PCR amplification using the primers BpfD-FlagKinF and BpfD-FlagKinR, and the yielding fragment was digested by *XbaI* and *XhoI*, defined as pK19-BpfD-Cter-*XbaI-XhoI*. A commercially synthesized nucleotide sequence GAT TAC AAG GAT GAC GAC GAT AAG GAC TAT AAG GAC GAT GAT GAC AAG GAC TAC AAA GAT GAT GAC GAT AAA encoding 3×Flag (DYKDDDDKDYKDDDDKDYKDDDDK) was used as a template and amplified by primers Flag-F/Flag*bpfD*-R. The yielding fragment was digested with *XbaI* and *XhoI*, which was cloned into pK19-BpfD-Cter-*XbaI-XhoI* to yield the BpfD C-terminal 3×Flag-tag knock-in plasmid, pK19-BpfD-Cter-FlagKin, which was conjugated into WT. Thus, the 3×Flag nucleotide sequence was inserted after *bpfD* in-frame by homologous recombination to yield WT/BpfD-Flag. The final transformant was verified by PCR, using the following primer pairs BpfD-CSF/BpfD-CSR, BpfD-COF/BpfD-CSR, and BpfD-CSF/BpfD-COR. A similar strategy was used to generate Δ*crp*/BpfD-Flag, Δ*cya*/BpfD-Flag, Δ*1291Δ3328*/BpfD-Flag, and Δ*crpΔ1291Δ3328*/BpfD-Flag. All transformants were verified by PCR and DNA sequencing. To construct a 3×Flag-tagged BpfA transformant, 3×Flag was inserted after residue 3700 aa (11100 bp) in the full-length protein of 4220 aa and the construction strategy was similar to that of WT/BpfD-Flag. The transformant was verified by PCR and DNA sequencing.

To construct a 1×HA-tagged BpfG transformant, HA was inserted after residue 221 aa (663 bp) in the full-length protein (235 aa)[64]. The construction strategy was as follows: the 3′-terminus region of *bpfG*, including its upstream and downstream flanks was amplified by the primers BpfG-C14-5F and BpfG-C14-3R and cloned into pK19*mobsacB* to yield an intermediate plasmid pK19-BpfG-Cter; pK19-BpfG-Cter was linearized by PCR amplification using the 5′-phosphorylated primers BpfG-HAKinF and BpfG-HAKinR; the linearized plasmid was self-ligated to yield the BpfG-HA knock-in plasmid, pK19-BpfG-C14-HAKin; and finally, the plasmid was conjugated into WT/BpfD-Flag to obtain the transformant WT/BpfD-Flag/BpfG-HA. The final transformants were verified by PCR, using the primers BpfG-C14-SF/BpfG-C14-SR, BpfD-C14-OF/BpfD-C14-SR, and BpfD-C14-SF/BpfD-C14-OR. A similar strategy was used to generate the transformants Δ*crp*/BpfD-Flag/BpfG-HA, Δ*cya*/BpfD-Flag/BpfG-HA, Δ*1291Δ3328*/BpfD-Flag/BpfG-HA, and Δ*crpΔ1291Δ3328*/BpfD-Flag/BpfG-HA. All transformants were verified by PCR and DNA sequencing.

To construct the 10×His-tagged CRP transformant, the 3′-terminus region of the *crp* gene, including its upstream and downstream flanks was amplified by primers CRP-C5F/CRP-C3R and cloned into pK19*mobsacB* to yield an intermediate plasmid pK19-CRP-Cter; pK19-CRP-Cter was linearized by PCR amplification using the primers CRP-His₁₀KinF and CRP-His₁₀KinR, both of which are 5′-phosphorylated primers containing the nucleotide sequence 10×CAT encoding 10×His. Next, the linearized plasmid was self-ligated to yield the CRP-His₁₀ knock-in plasmid, pK19-CRP-C-His₁₀Kin, which was conjugated into WT/BpfD-Flag/BpfG-HA to obtain the transformant WT/BpfD-Flag/BpfG-HA/CRP-His. The final transformants were verified by PCR, using the primer pairs CRP-CSF/CRP-CSR, CRP-COF/CRP-CSR, and CRP-CSF/CRP-COR. A similar strategy was used to generate Δ*cya*/BpfD-Flag/BpfG-HA/CRP-His. All transformants were verified by PCR and DNA sequencing.

To replace the *bpfA* operon promoter (P$_{bpfA}$) with the constitutive *aacC1* promoter[77], P$_{bpfA}$ and its flanking regions were amplified by the primers P$_{bpfA}$-5F/P$_{bpfA}$-3R and cloned into pK19*mobsacB* to yield an intermediate plasmid pK19-P$_{bpfA}$UD. The intermediate plasmid was then used as a template to amplify the P$_{bpfA}$-free part with the primers P$_{bpfA-aacC1}$-KinF and P$_{bpfA-aacC1}$-KinR, yielding a linearized plasmid pK19-UD. The *aacC1* promoter region (P$_{aacC1}$) was amplified using the plasmid pUCGm as a template with the primers P$_{aacC1}$-F and P$_{aacC1}$-R. Then, the P$_{aacC1}$ was ligated to pK19-UD to yield pK19-P$_{aacC1}$-UD, which was conjugated into the WT. Thus, P$_{aacC1}$ was inserted and replaced P$_{bpfA}$ by homologous recombination, yielding WT/P$_{aacC1}$-*bpfA*. The transformant was verified by PCR and DNA sequencing.

In order to analyze whether cAMP influences the interaction between CRP and BpfD, *crp*-R84L-His₁₀ was knocked into Δ*crp*/P$_{aacC1}$-*bpfA*/BpfD-Flag/BpfG-HA to yield WT/P$_{aacC1}$-*bpfA*/BpfD-Flag/BpfG-HA/CRP-R84L-His. The binding ability between CRP and BpfD was compared to CRP-R84L and BpfD using a Co-IP assay.

**Biofilm microtiter plate assay**. Overnight cultured LB seed broth was diluted to OD$_{600}$~0.01 using MM1 medium, and 100 μl of diluent was aliquoted into 96-well cell culture plates (NEST, China). When necessary, 1 mM exogenous adenosine 3′, 5′-cyclic monophosphate sodium salt monohydrate (cAMP) (Sigma-Aldrich, USA) was added into MM1 medium. The 96-well cell culture plates were statically incubated at 30 °C for different times. The biofilm assay was performed based on a crystal violet dyeing method[51]. Specifically, after incubation, the planktonic cells were removed and the surface attached cells were washed twice using deionized water. Each well was added with 150 μl of 1% crystal violet solution and incubated for 15 min. The crystal violet solution was removed and the wells were washed twice with 200 μl of deionized water. Then, 200 ul of 95% ethanol solution was added to each well, and the absorbance at 570 nm was measured using a microplate reader (ELx800, BioTek, USA) to determine the biofilm biomass.

**RNA extraction and real-time RT-PCR (qRT-PCR) assay**. *S. putrefaciens* CN32 cells grown in 96-well plates were harvested at appropriate time points for RNA extraction using the TRIzol method. Next, 2 μg total RNA was reverse transcribed to cDNA following the manufacturer's protocol (Promega, USA), and cDNA was used as template for the qRT-PCR analysis. The qRT-PCR assay was performed using the Power SYBR$^{TM}$ Green PCR mix (Applied Biosystems, USA) and analyzed using the BioRad CFX96 Touch System. The PCR program included a pre-denaturation step at 95 °C for 10 min, 40 cycles of 95 °C for 10 s and 60 °C for 30 s; the fluorescence was measured at the end of each cycle. The primers used in qRT-PCR analysis were listed in Supplementary Table 3, and the 16S rRNA gene was selected as an internal control. All of the experiments were performed at least three times.

**Protein purification**. To purify His₆-CRP and His₆-CRP-R84L, pET28a-CRP and pET28a-CRP-R84L were constructed and introduced into *E. coli* BL21 (DE3) cells, which were cultured at 37 °C in LB medium to an OD$_{600}$~0.6 and were induced with 0.4 mM isopropyl-β-D-thiogalactopyranoside (IPTG) for 24 h at 16 °C, and the His₆-CRP and His₆-CRP-R84L proteins were purified by Ni-agarose resin (CoWin Biosciences, China) according to the manufacturer's protocol.

To purify GST-BpfD-In (intracellular domains), pGEX-4T-1-BpfD-In was constructed and introduced into *E. coli* BL21 (DE3) cells, which were cultured at 37 °C in LB medium to an OD$_{600}$~0.6. The cells were then inducted with 0.4 mM IPTG for 24 h at 16 °C, and the GST-BpfD intracellular domains were purified by Glutathione-Sepharose resin (Solarbio Life Sciences, China) according to the manufacturer's protocol. Meanwhile, the GST protein was purified.

To purify MBP-1291 and MBP-1291(GGAAF), pMAL-c2x-1291 and pMAL-c2x-1291-GGAAF, were constructed and introduced into *E. coli* BL21 (DE3) cells, respectively, which were cultured at 37 °C in LB medium to an OD$_{600}$~0.8 and induced with 0.6 mM IPTG for 20 h at 16 °C, and the harvested cells were resuspended in MBP binding buffer (20 mM Tris-HCl, pH 7.4, 200 mM NaCl, 1 mM EDTA). Proteins were purified with PurKine$^{TM}$ MBP-tag dextrin resin (Abbkine Scientific, China) using elution buffer (20 mM Tris-HCl, pH 7.4, 1 mM EDTA, 10 mM maltose).

**Electrophoretic mobility shift assay (EMSA)**. To perform EMSA experiment, the DNA probe *bpfA*-pro covered the *bpfA* promoter region from −219 bp to +11 bp relative to the *bpfA* start codon was amplified, and the purified probe was labeled with digoxigenin (DIG) using the DIG Gel Shift Kit, 2nd Generation (Roche, USA). EMSA was performed according to the manufacturer's protocol. When necessary, 1 μM cAMP was added. The primers used to amplify the EMSA probes are listed in Supplementary Table 3.

**cAMP measurement**. The intracellular cAMP concentration was measured using an established method[78]. Specifically, *S. putrefaciens* CN32 cells were cultured in 96-well plates. When necessary, 1 mM exogenous cAMP (Sigma-Aldrich, USA) was added to MM1 medium. Cells were harvested at 30 h by centrifugation at 13,000 *g* for 2 min at 4 °C and washed twice with pre-cooled phosphate-buffered saline (PBS) buffer. Then, the cell samples were divided into two parts: one part was used to determine the total protein concentration using a Quick Start Bradford 1×dye reagent (BioRad, USA), and the other was acetylated following the manufacturer's protocol and used to measure the intracellular cAMP concentration with a Cyclic AMP ELISA Kit (Cayman Chemical, USA). The intracellular cAMP concentrations were converted to picomoles per milligram of protein.

**c-di-GMP measurement**. *S. putrefaciens* CN32 cells grown in 96-well plate were harvested at 30 h by centrifugation at 13,000 *g* for 2 min at 4 °C and washed twice with precooled PBS buffer. Then, the cell samples were divided into two parts: one part was used to determine the total protein concentration using a Quick Start Bradford 1×dye reagent (BioRad, USA); the other was lysed by B-PER Bacterial Protein Extraction Reagent (Thermo Fisher Scientific, USA), incubated at room temperature for 10 min, and then centrifuged at 13,000 *g* for 5 min. The liquid supernatant was used to measure the intracellular c-di-GMP concentration using a Cyclic di-GMP ELISA Kit (Cayman Chemical, USA). The intracellular c-di-GMP concentrations were converted to picomoles per milligram of protein.

**DGC activity assays in vitro**. The DGC activity was analyzed according to an established method[79]. Specifically, Sputcn32_1291 and GTP were dissolved using the reaction buffer (50 mM Tris-HCl, pH 7.6, 10 mM MgCl₂, 0.5 mM EDTA, 50 mM NaCl) to 10 μM and 300 μM, respectively. Then, 100 μl of Sputcn32_1291 solution was mixed with 300 μl of GTP solution, and incubated at 30 °C for 5 h. Subsequently, the reaction mixture was filtered through a 0.22 μm filter, and the filtrates were analyzed using a reversed-phase HPLC (EClassical 3100 HPLC system, Elite, China) equipped with a C18 column (Supersil ODS2, 5 μm, 4.6 × 200 mm, Elite, China). Buffer A (100 mM KH₂PO₄, 4 mM tetrabutyl ammonium hydrogen sulfate, pH 5.9) and buffer B (75% buffer A, 25% methanol) were used for product separation in a gradient program (minute and buffer B percentage): 0.0, 0%; 2.5, 0%; 5.0, 30%; 10.0, 60%; 14.0, 100%; 21.0, 100%; 22.0, 50%; 23.0, 0% and 30.0, 0% at 40 °C with a flow rate of 0.7 ml min$^{-1}$. Nucleotides were detected at a wavelength of 254 nm.

**BpfA localization assay**. The cells of different strains grown in 96-well plates were harvested and adjusted to the same cell concentration. Briefly, 30 ml of adjusted cells was harvested by centrifugation at 13,000 $g$ for 5 min at 4 °C, resuspended in 150 μl PBS, and mixed with 150 μl PBS containing 8 mg ml$^{-1}$ lysozyme (Solarbio Life Sciences, China). The obtained cell suspensions were incubated at 37 °C for 15 min, and then centrifuged again (13,000 $g$, 4 °C, 5 min). Finally, the BpfA-containing supernatant fraction was assayed by western blotting and dot blotting.

**Dot blotting**. Dot blotting was performed using established method[80,81]. Specifically, the 10 μl sample of BpfA on the cell surface from different strains was dropped on a PVDF membrane (Roche Diagnostics, Germany), dried at 37 °C for 60 min, and incubated at 4 °C overnight. Flag-tagged BpfA was detected by western blotting analysis.

**GST pull-down experiment**. Proteins used for GST pull-down assay were heterologously expressed and purified. His$_6$-CRP, His$_6$-CRP-R84L, GST-BpfD-In, and GST protein were dialyzed with PBS-glycerol buffer (PBS buffer containing 20% of glycerol). Equal amounts of GST-tagged and His$_6$-tagged proteins were mixed, and 20 μM cAMP was supplied if necessary. Pull-down buffer (50 mM Na$_2$HPO$_4$/NaH$_2$PO$_4$, pH 7.8, 200 mM NaCl, 1 mM EDTA, 0.5% NP-40) was added to a final volume of 1 ml. Subsequently, 50 μl pre-balanced Glutathione-Sepharose Resin (Solarbio Life Sciences, China) was supplied to each volume. Following a 2 h incubation with generous rotation at 4 °C, the Glutathione-Sepharose Resin was washed five times with pull-down buffer, and 20 μM cAMP was supplied if necessary. The bound proteins were analyzed by western blotting.

**Co-immunoprecipitation (Co-IP) assay**. Co-IP was performed to verify the interaction between Flag-tagged BpfD and His-tagged CRP and the interaction between Flag-tagged BpfD and HA-tagged BpfG. Strains grown in 96-well plates at 30 h were harvested, washed once with PBS, and resuspended in ice-cold lysis buffer (45 mM HEPES, pH 7.2, 10% glycerol, 0.2% NP-40, 150 mM NaCl, 1 mM EDTA, 2 mM DTT, and 1×protease inhibitor cocktail [CoWin Biosciences, China]) for 30 min. The cell lysates were clarified by centrifugation at 4 °C, 13,000 $g$ for 20 min. The total protein concentration in the supernatant was determined by a Quick Start Bradford 1×dye reagent (BioRad, USA) and adjusted to 2 mg ml$^{-1}$. The supernatant was incubated with monoclonal anti-Flag M2 antibody produced in mouse (Sigma-Aldrich, USA) and GammaBind G Sepharose beads (GE Healthcare, USA) on a rotary shaker at 4 °C for 2 h. The IgG from mouse serum (Sigma-Aldrich, USA) was used as negative control. Then, the protein-bead complexes were washed using lysis buffer for three times. When necessary, 5 μM c-di-GMP was added to the lysis buffer to perform interaction analysis between Flag-tagged BpfD and HA-tagged BpfG[64], and 5 μM cAMP was added to the lysis buffer to perform interaction analysis between Flag-tagged BpfD and His-tagged CRP. Then, the bound protein complexes were eluted from beads using sample loading buffer and analyzed by western blotting.

**Biotinylated cAMP pull-down assay**. The binding between cAMP and CRP and CRP-R84L was analyzed using Dynabeads™ M-280 Streptavidin (Thermo Fisher Scientific, USA). The streptavidin magnetic beads were washed by TTBS (0.1 M Tris-HCl, pH 8.0, 0.9% (w/v) NaCl, 0.1% (v/v) Tween-20) before use. 2 nmol Biotin-cAMP conjugate (AAT Bioquest, USA) and equal amount of protein (His$_6$-CRP or His$_6$-CRP-R84L) were incubated in reaction buffer (10 mM Tris-HCl, pH 7.5, 50 mM KCl, 1 mM DTT) at 30 °C for 30 min. 2 nmol cAMP (Sigma-Aldrich, USA) or Biotin (Sigma-Aldrich, USA) were incubated with protein as controls. Half of the mixture was kept as input, while the other half was combined with the washed beads and incubated for 1.5 h at room temperature on a rotary shaker. The beads were collected using a magnetic stand and the supernatant was removed. After washing the beads using TTBS for three times, the beads were added with sample loading buffer and boiled to elute the bound protein, and the protein was then analyzed by western blotting.

**Extraction of intracellular total proteins**. In order to analyze the intracellular protein levels of Sputcn32_1291, Sputcn32_3328, and BpfA, the total proteins of *S. putrefaciens* CN32 cells collected from 96-well plates were extracted using B-PER Bacterial Protein Extraction Reagent (Thermo Fisher Scientific, USA); and the samples were adjusted to an equal amount of protein for western blotting assay.

**Western blotting**. The protein samples were separated by SDS-PAGE, and then transferred onto a PVDF membrane (Roche Diagnostics, Germany). After blocking with skim milk (5% in TBST), the membranes were incubated with primary antibody (Monoclonal anti-Flag M2 antibody produced in mouse, Sigma-Aldrich, USA; Anti HA-Tag mouse monoclonal antibody, Anti His-Tag mouse monoclonal antibody and Anti GST-Tag mouse monoclonal antibody were purchased from CoWin Biosciences, China) against the target protein was supplied, followed by incubation with Goat anti-mouse IgG HRP conjugated secondary antibody (CoWin Biosciences, China), or Goat Anti-Mouse IgG-Fc HRP conjugated secondary antibody (Sino Biological, China) for Co-IP experiments. The target protein was then detected using the eECL Western Blotting Kit (CoWin Biosciences, China) according to the manufacturer's protocol.

**Microscale thermophoresis (MST) assay**. Microscale thermophoresis (MST) was performed to qualitatively analyze the interaction between BpfD-In and CRP. Proteins were heterologously expressed and purified. His$_6$-CRP was dialyzed with PBS-glycerol buffer. GST and GST-BpfD-In were freeze-dried after dialysis with deionized water. Subsequently, 100 μl of 10 μM His$_6$-CRP was labeled with NHS NT-647 dye using the Red-NHS 2nd Generation Labeling Kit (NanoTemper, Germany), which was eluted with a reaction buffer (PBS-glycerol buffer: Pull-down buffer = 1:1). Next, 10 μl of labeled His$_6$-CRP was mixed with 10 μl of 5 μM GST or GST-BpfD-In or with the same volume of reaction buffer, respectively. Finally, samples were loaded into capillaries and analyzed using a Monolith Instrument NT.115 (NanoTemper, Germany) in Binding Check mode. Both the LED power and MST power were set to 20%.

**Statistics and reproducibility**. Statistical analyses were performed using Graph-Pad Prism (version 9.0.0). All experiments were performed at least three independent times. Data are presented as a mean ± SD (standard deviation). Statistical significance was determined using two-sided Student's $t$ test. $p$ values are reported using the following symbolic representation: NS (No significance) $p > 0.05$, $*p < 0.05$, $**p < 0.01$, $***p < 0.001$.

**Reporting summary**. Further information on research design is available in the Nature Research Reporting Summary linked to this article.

## Data availability

Source Data are provided with this paper. All the data supporting this study are available in the main article, Supplementary Information files, Source Data file, or from the corresponding authors upon request. Source data are provided with this paper.

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

## Acknowledgements

We thank Professor Guoliang Qian for gifting the strain *E. coli* MG1655. This study is supported by National Natural Science Foundation of China (31970036, 31900401 and 31800020), Natural Science Foundation of the Jiangsu Higher Education Institutions of China (20KJB180001, 20KJA180007); Natural Science Foundation of Jiangsu Province (BK20181009, BK20210920), Natural Science Foundation of Xuzhou city (KC19196), Six Talent Peaks Project of Jiangsu Province (JNHB-103), Priority Academic Program Development of Jiangsu Higher Education Institutions.

## Author contributions

C.L., D.S., and W.J.L. planed and designed researches. C.L., Y.C., X.G.Z., Y.R.R., and J.R.Z. carried out the mutant constructions and biofilm assays. C.L., W.J.L., and D.S. performed blotting experiments, c-di-GMP and cAMP measurements. J.W.L. and D.S. carried out MST assay. C.L. and D.S. analyzed the data. W.J.L., C.L., and D.S. wrote the paper and all authors revised it.

## Competing interests

The authors declare no competing interests.
