## [Peer Review File · Nature Communications]

Reviewers' Comments:

Reviewer #1:

Remarks to the Author:

The manuscript entitled "cAMP and c-di-GMP synergistically support biofilm maintenance through the direct interaction of their effectors" by Liu et al., studies the role of CRP-cAMP in the maintenance of biofilms in *Shewanella putrefaciens* CN32. The bulk of studies involved deletions of genes associated with CRP, cAMP production and degradation, adhesins (BpfA), among many others, to determine the genetic factors involved in biofilm maintenance. In addition to the genetic studies, the authors also include protein-protein interaction using co-IP, MST. Altogether, this is a comprehensive and detailed study, wherein the authors find that CRP-cAMP and c-di-GMP support biofilm maintenance through retaining BpfA on the cell surface, rather than through regulating bpfA transcription.

While this study is thorough, and the findings are of interest since they provide a mechanism by which CRP-cAMP regulate a non-transcription function, this reviewer is not convinced that the study is novel enough for a broad readership like *Nat Commun*. The authors state in many parts of their manuscript that this is the first ever description of a non-transcription regulation function of CRP. However, I disagree and the authors themselves provide references (some but not all) of previous studies where CRP from bacteria play roles that are not only of transcription of genes. For instance, others have provided direct and indirect evidence that CRP in *M. tuberculosis* has other roles apart from regulating transcription of genes, such as a scaffolding protein of chromosome organization.

a. Knapp GS, Lyubetskaya A, Peterson MW, Gomes AL, Ma Z, Galagan JE, McDonough KA. Role of intragenic binding of cAMP responsive protein (CRP) in regulation of the succinate dehydrogenase genes Rv0249c-Rv0247c in TB complex mycobacteria. *Nucleic Acids Res*. 2015 Jun 23;43(11):5377-93. doi: 10.1093/nar/gkv420. Epub 2015 May 4. PMID: 25940627; PMCID: PMC4477654.

b. Kahramanoglou C, Cortes T, Matange N, Hunt DM, Visweswariah SS, Young DB, Buxton RS. Genomic mapping of cAMP receptor protein (CRP Mt) in *Mycobacterium tuberculosis*: relation to transcriptional start sites and the role of CRPMt as a transcription factor. *Nucleic Acids Res*. 2014 Jul;42(13):8320-9. doi: 10.1093/nar/gku548. Epub 2014 Jun 23. PMID: 24957601; PMCID: PMC4117774.

c. Gárate F, Dokas S, Lanfranco MF, Canavan C, Wang I, Correia JJ, Maillard RA. cAMP is an allosteric modulator of DNA-binding specificity in the cAMP receptor protein from *Mycobacterium tuberculosis*. *J Biol Chem*. 2021 Jan-Jun;296:100480. doi: 10.1016/j.jbc.2021.100480. Epub 2021 Feb 26. PMID: 33640453; PMCID: PMC8026907.

d. Review: Johnson RM, McDonough KA. Cyclic nucleotide signaling in *Mycobacterium tuberculosis*: an expanding repertoire. *Pathog Dis*. 2018 Jul 1;76(5):fty048. doi: 10.1093/femspd/fty048. PMID: 29905867; PMCID: PMC6693379.

Similarly, the sentence "all subsequent studies focused on CRP-cAMP as a transcription factor, and no previous study has reported that CRP-cAMP regulates physiological metabolism as a non-transcriptional regulator" (sentence 72-73) is not correct. Sentence 123-126 is also misleading in that this study is not the first study showing CRP-cAMP acting as a non-transcriptional regular. The authors themselves describe in the introduction (sentences 123-126) previous studies and reviews showing that CRP-cAMP is involved in biofilm formation. What seems to be unknown is the mechanism, which is what this study begins to address.

In my opinion, sentences 72-73, 123-126 as well as the abstract sentence 34 need to be modified to acknowledge other studies that show other functions that are non-strictly transcription of genes. This study goes aims to address the mechanism of involvement of CRP-cAMP in biofilms. Therefore, I believe the results belong to a more specialized readership.

In addition, I have concerns of how the authors cite or refer the results from CRP-Ecoli in a general way. Sentences 58-66 in the Introduction refer to CRP from *E. coli* and it must be stated as such. CRP proteins from other bacteria have been studied, and some examples are provided by the authors. But the mechanisms and biophysical and structural studies in the first sentences of the introduction refer to the *E. coli* homolog. For instance, others have shown biophysically that

binding to promoter sequences is cAMP independent for *M. tuberculosis*. For instance:

a. Stapleton M, Haq I, Hunt DM, Arnvig KB, Artymiuk PJ, Buxton RS, Green J. Mycobacterium tuberculosis cAMP receptor protein (Rv3676) differs from the Escherichia coli paradigm in its cAMP binding and DNA binding properties and transcription activation properties. *J Biol Chem*. 2010 Mar 5;285(10):7016-27. doi: 10.1074/jbc.M109.047720. Epub 2009 Dec 22. PMID: 20028978; PMCID: PMC2844151.

b. Gárate F, Dokas S, Lanfranco MF, Canavan C, Wang I, Correia JJ, Maillard RA. cAMP is an allosteric modulator of DNA-binding specificity in the cAMP receptor protein from Mycobacterium tuberculosis. *J Biol Chem*. 2021 Jan-Jun;296:100480. doi: 10.1016/j.jbc.2021.100480. Epub 2021 Feb 26. PMID: 33640453; PMCID: PMC8026907.

I also have concerns on the results from the mutation study using CRP-R84L. The authors claim that this mutant abolishes cAMP activity. Did the authors measure this? I couldn't find it. They cite reference #3 but this is for *E. coli*. While the position seems to be conserved, it is important that the authors show that there is no detectable cAMP binding in order to reach the conclusion that "the ability of CRP to enhance the interaction between BpFD and BpFG is lost in the absence of cAMP"

Lastly, the authors seem to claim in the Discussion that their findings may apply or may be more ubiquitous in bacteria. They state that heterologous complementation can partially restore biofilm maintenance with CRP from *E. coli* in *Deltacrpdeltacrpa* but not in *Deltacrpdeltacrya* but their data is not shown? This is a major claim of this article and the authors must show those results to support their claim that CRP-cAMP may play in biofilm in most bacteria (i.e., ubiquitous).

Altogether, the study is comprehensive and detailed (somewhat difficult to read too given the specialized terminology) but some assays and measurements are needed. But overall the impact and novelty of the work will be better suited in a more specialized journal.

Reviewer #2:

Remarks to the Author:

Liu and colleagues present a voluminous amount of work using complementary techniques from molecular genetics, biochemistry and molecular biology supporting a regulatory mechanism whereby cAMP-CRP interacts with a membrane-bound receptor to affect biofilm formation in microplates. The regulatory mechanism reported in the paper represents a convergence point for cAMP and c-di-GMP signaling. Many reports in the field recognize cross-talk between these second messengers yet mechanisms explaining the connections remain scant. The discovery that the cAMP-CRP transcription factor post-translationally regulates a receptor complex important for processing an adhesin is highly novel and will be of interest to many in the field. The work appears technically sound with only a couple of controls that appear to be missing (outlined below). However, the work would benefit greatly from editing and a key shortcoming, which is addressable, is a lack of integration with terminology describing the biofilm ecological cycle.

Major points of constructive feedback

A major constructive criticism for the authors is that the term biofilm "maintenance" is not integrated within the context of the biofilm ecological cycle that is summarized succinctly on lines 79-80. At what stage of this ecological cycle is Crp exerting an effect? Is the cAMP-CRP complex actively controlling a process of biofilm dispersion? That's the case with the Lap system in *Pseudomonas*, which the authors contrast to the BpgAGD system. The Lap system regulates biofilm dispersal in response to phosphate starvation (see <https://www.pnas.org/content/106/9/3461>). With respect to the writing, the work would benefit greatly from better integration with existing terminology and concepts used in the field.

With respect to better integration with the biofilm ecological cycle, can the authors provide time-dependent microscopy data to look at what's happening in their CRP mutant? For example, one possible interpretation is that cAMP-CRP suppresses nutrient-dependent biofilm dispersion. Another possibility is that the cells die (such is the case with another study looking at biofilm

"maintenance," <https://pubmed.ncbi.nlm.nih.gov/33531388/>). These outcomes could be evaluated using one of several continuous or batch-culture biofilm models amenable to microscopy. This does not need to be a major undertaking to provide this key information.

A tremendous amount of work has gone into generating deletion mutants and plasmids for expressing multiple genes simultaneously; however, the vector controls used in the complementation analyses (e.g. for the expression of *crp*, *cya*, and *cpdA*) are missing.

Lines 211-289. It is very difficult to follow the logic in this section because of the way it is written. I'm also unsure whether all this work is necessary for this one manuscript and whether communication might be better if some of the information and data were removed. It's uncanny that CRP regulates the transcription of so many genes for DGCs and PDEs! It would be great if the authors could home in on just the results that lead the reader to conclude that cAMP-CRP affects biofilm formation independently of the regulation of all these genes. Also, it isn't clear why a handful of DGC proteins would need to be produced, purified and verified for activity because of the pleiotropic effects Crp has on gene regulation. Can the authors trim down the data to the key dataset(s) and simplify the presentation?

Minor points

Lines 93-108. A comparison is made between the BpfAGD systems and the *Pseudomonas* Lap system. Are the systems orthologous, or is this comparison simply analogous? Could the authors provide more detail about the molecular functioning of the systems and clarify, please? As it pertains to this point, Supplementary Figure 1 and Figure 6 are redundant. The two could be combined. I would suggest presenting the model as Figure 1 in the paper because basic information about the BpfAGD system is critical for understanding the content in the paper. Presenting a revised model up front will allow the readers to better follow along, even if the cAMP-CRP interaction with BpfD, which is the major contribution forthcoming from the present manuscript, is presented first. Also – aren't there 4 outcomes from the model and not just two (because there are 2 second messengers at play, both of which might be high or low)?

Lines 116 and 371 (and elsewhere). "...bacterial bistable transition." Bistability refers to a very specific type of gene expression, and this term is not used correctly in the context of this paper. Do the authors mean "the transition between motile planktonic and sessile biofilm lifestyles in many bacteria?"

Lines 132-134. If the bacteria are still growing, they are not in stationary phase. Do the authors mean that the *crp* mutant grew to a higher cell density than the other strains?

Line 136 (and elsewhere). Remove "obvious" and replace with "statistically significant."

Line 142. "tertiary-knockout strain." I recommend using the language of geneticists, which would be to state "a strain lacking all 3 adenylate cyclases." Similarly, "tertiary-complementation strain" should be a "strain bearing a plasmid encoding all 3 adenylate cyclases." Some additional details added to the results section would better help the reader to understand how these constructs were built.

Lines 172-174 (figure 2B). How do the authors know that cAMP isn't carried over from media to the nucleotide extractions? I'm not sure if that matters – but wording around presentation of the data could be more careful.

Line 200 (Figure 2J). There are multiple numbers that appear in Fig. 2J with respect to genotypes and these are not explained sufficiently in the text when the data are first presented.

Reviewer #3:

Remarks to the Author:

In this manuscript the authors investigate aspects of the molecular mechanisms of biofilm formation mediated by the cyclic AMP second messenger signaling system in *Shewanella putrefaciens* CN32. They could show that the cyclic AMP receptor protein CRP in complex with its

ligand cAMP, a transcriptional regulator, regulates maintenance of biofilm formation in the late phase of growth upon incubation under steady state conditions not only by transcriptional regulation of cyclic di-GMP turnover proteins, but also by direct interaction with the cyclic di-GMP receptor BpfD. BpfD, upon cyclic di-GMP binding and interaction with the cAMP-loaded CRP, prevents the protease BpfG to cleave the large surface protein BpfA in order to dissolve the biofilm.

This work adds a major novel aspect to the complex regulation of biofilm formation in *S. putrefaciens* and the interplay between components of two ubiquitous second messenger signaling systems. Furthermore, the authors show that the molecular mechanisms of regulation of this work potentially extend to other bacterial species, in particular *Escherichia coli*. The work is presented very organized and overall clearly presented. It would be however highly beneficial to strengthen the findings by identification of amino acids in CRP which do not promote interaction with BpfD. As *S. putrefaciens* has three adenylate cyclases, it would be relevant to know whether all three enzymes contribute equally to biofilm maintenance in the stationary phase, or whether one of those enzymes has a dominant role. Does any of those proteins contain an N-terminal signaling domain that allows conclusions about a specific activation in the stationary phase of growth? Perhaps this part of the signaling pathway in combination with the cyclic di-GMP turnover proteins that are regulated by cAMP-CRP can be included in the figure.

Specific comments:

I.32: ...the complex cross-regulation of various second messengers has been hardly explored.

I.33: textbook? ..one of the most well investigated transcription factors.

I.34: ...act not only as a transcription factor, but also through interaction with an effector protein.

I.36: ...to regulate proteolytic cleavage of BfpA by BpfG.

I.125: ...acting not only as a transcriptional regulator by DNA binding, but also acting through protein-protein interactions with a cyclic di-GMP receptor to modulate effector efficiency.

I.154: ...we deleted...

I.189: As the authors point out later in the manuscript, not necessarily, in particular also as cAMP-CRP as a transcription factor affects transcription of hundreds of genes that might also be involved in the process of surface exposure of BpfA

I.234: Besides the full domain structure, can the authors comment whether and which domains are expected to be functional for 1291 and 3328,

I.267 and elsewhere: Not really, as cAMP-CRP positively regulates the transcription of the bpfAGD operon.

I.323: ...integration of a 3XFLAG-tag nucleotide sequence 5' of the stop-codon of...

I.452: ---by bacteria themselves...

I.495: ...was introduced into...

I.504: The nucleotide sequences corresponding to tags were inserted upstream of the...

I.516: ..inserted or integrated...

References for cyclic di-GMP signaling are highly biased. The authors might want to consider to include references the report the original finding that biofilm formation is positively regulated by cyclic di-GMP in a number of organisms.

Point to point response to REVIEWER COMMENTS:

Reviewer #1 (Remarks to the Author):

The manuscript entitled “cAMP and c-di-GMP synergistically support biofilm maintenance through the direct interaction of their effectors” by Liu et al., studies the role of CRP-cAMP in the maintenance of biofilms in *Shewanella putrefaciens* CN32. The bulk of studies involved deletions of genes associated with CRP, cAMP production and degradation, adhesins (BpfA), among many others, to determine the genetic factors involved in biofilm maintenance. In addition to the genetic studies, the authors also include protein-protein interaction using co-IP, MST. Altogether, this is a comprehensive and detailed study, wherein the authors find that CRP-cAMP and c-di-GMP support biofilm maintenance through retaining BpfA on the cell surface, rather than through regulating bpfA transcription.

While this study is thorough, and the findings are of interest since they provide a mechanism by which CRP-cAMP regulate a non-transcription function, this reviewer is not convinced that the study is novel enough for a broad readership like *Nat Commun*. The authors state in many parts of their manuscript that this is the first ever description of a non-transcription regulation function of CRP. However, I disagree and the authors themselves provide references (some but not all) of previous studies where CRP from bacteria play roles that are not only of transcription of genes. For instance, others have provided direct and indirect evidence that CRP in *M. tuberculosis* has other roles apart from regulating transcription of genes, such as a scaffolding protein of chromosome organization.

a. Knapp GS, Lyubetskaya A, Peterson MW, Gomes AL, Ma Z, Galagan JE, McDonough KA. Role of intragenic binding of cAMP responsive protein (CRP) in regulation of the succinate dehydrogenase genes Rv0249c-Rv0247c in TB complex mycobacteria. *Nucleic Acids Res.* 2015 Jun 23;43(11):5377-93. doi: 10.1093/nar/gkv420. Epub 2015 May 4. PMID: 25940627; PMCID: PMC4477654.

b. Kahramanoglou C, Cortes T, Matange N, Hunt DM, Visweswariah SS, Young DB, Buxton RS. Genomic mapping of cAMP receptor protein (CRP Mt) in *Mycobacterium tuberculosis*: relation to transcriptional start sites and the role of CRPMt as a transcription factor. *Nucleic Acids Res.* 2014 Jul;42(13):8320-9. doi: 10.1093/nar/gku548. Epub 2014 Jun 23. PMID: 24957601; PMCID: PMC4117774.

c. Gárate F, Dokas S, Lanfranco MF, Canavan C, Wang I, Correia JJ, Maillard RA. cAMP is an allosteric modulator of DNA-binding specificity in the cAMP receptor protein from *Mycobacterium tuberculosis*. *J Biol Chem*. 2021 Jan-Jun;296:100480. doi: 10.1016/j.jbc.2021.100480. Epub 2021 Feb 26. PMID: 33640453; PMCID: PMC8026907.

d. Review: Johnson RM, McDonough KA. Cyclic nucleotide signaling in *Mycobacterium tuberculosis*: an expanding repertoire. *Pathog Dis*. 2018 Jul 1;76(5):fty048. doi: 10.1093/femspd/fty048. PMID: 29905867; PMCID: PMC6693379.

Reply: We thank the reviewer #1 for his professional comments and very good suggestions. The cAMP receptor protein (CRP) is ubiquitous in bacteria. The complex cross-regulation of various second messengers has been hardly explored. The main innovation of this study is the discovery of a new regulation model about cAMP and c-di-GMP synergistically support biofilm maintenance through the direct interaction of their effectors. In previous studies, CRP, as an important ubiquitous protein in bacteria, was mainly studied as a transcription factor. We agree with the comments of reviewer 1 that some studies have provided direct and indirect evidence that CRP in *M. tuberculosis* has other roles apart from regulating transcription of genes, such as a scaffolding protein of chromosome organization. In this study, we found that CRP regulates physiological metabolism through protein-protein interaction, which is independent of DNA.

We thank reviewer 1 for his useful comments and recommended references. In the revised manuscript, we added new information about the role of CRP as a non-transcription factor in *M. tuberculosis* to the revised manuscript as follows: “Although most subsequent studies focused on CRP-cAMP as a transcription factor, CRP-cAMP may play a non-transcriptional regulatory role in some bacteria. For instance, in *Mycobacterium tuberculosis*, CRP may act as a nucleoid associated protein (NAP) to influence the dynamic spatial arrangement of the chromosome in a cAMP-independent manner. However, in γ -proteobacteria, the researches on CRP-cAMP as a non-transcriptional regulator involved in physiological metabolism are limited”. Please see line 71 to line 76 in the revised manuscript. And we added three new references, please see new reference 16, 17, and 18 in the revised manuscript. In addition, we removed all the sentence “for the first time” in the revised manuscript.

Similarly, the sentence “all subsequent studies focused on CRP-cAMP as a transcription factor, and no previous study has reported that CRP-cAMP regulates physiological metabolism as a non-transcriptional regulator” (sentence 72-73) is not correct. Sentence 123-126 is also misleading in that this study is not the first study showing CRP-cAMP acting as a non-transcriptional regular. The authors themselves describe in the introduction (sentences 123-126) previous studies and reviews showing that CRP-cAMP is involved in biofilm formation. What seems to be unknown is the mechanism, which is what this study begins to address.

Reply: Thanks for this valuable suggestion. In the revised manuscript, we deleted the sentence “all subsequent studies focused on CRP-cAMP as a transcription factor, and no previous study has reported that CRP-cAMP regulates physiological metabolism as a non-transcriptional regulator”, and we added some updated information of the role of CRP-cAMP in *Mycobacterium tuberculosis* “Although most subsequent studies focused on CRP-cAMP as a transcription factor, CRP-cAMP may play a non-transcriptional regulatory role in some bacteria. For instance, in *Mycobacterium tuberculosis*, CRP may act as a nucleoid associated protein (NAP) to influence the dynamic spatial arrangement of the chromosome in a cAMP-independent manner. However, in γ -proteobacteria, the researches on CRP-cAMP as a non-transcriptional regulator involved in physiological metabolism are limited”. Please see line 71 to line 76 in the revised manuscript.

We deleted the sentence in original manuscript “but also describes for the first time that CRP-cAMP, acting as a non-transcriptional regulator, modulate a non-transcriptional-associated signal transduction pathway”, and it was changed into “but also describes a regulatory pattern that CRP-cAMP modulates biofilm maintenance acting as a post-translation regulator.”. Please see line 126 to line 127 in the revised manuscript.

In my opinion, sentences 72-73, 123-126 as well as the abstract sentence 34 need to be modified to acknowledge other studies that show other functions that are non-strictly transcription of genes. This study goes aims to address the mechanism of involvement of CRP-cAMP in biofilms. Therefore, I believe the results belong to a more specialized readership.

Reply: Thanks for pointing this out. In the revised manuscript, we added new information as following: “Although most subsequent studies focused on CRP-cAMP as a transcription factor, CRP-cAMP may play a non-transcriptional regulatory role in some bacteria. For instance, in

Mycobacterium tuberculosis, CRP may act as a nucleoid associated protein (NAP) to influence the dynamic spatial arrangement of the chromosome in a cAMP-independent manner. However, in γ -proteobacteria, the researches on CRP-cAMP as a non-transcriptional regulator involved in physiological metabolism are limited". Please see line 71 to line 76 in the revised manuscript. And we added new reference 16, 17, and 18 in the revised manuscript.

The sentence in abstract was changed into "The cAMP receptor protein (CRP) is a well-characterized global transcription factor, but investigations on CRP-cAMP as a post-translation regulator are limited" Please see line 32 to line 34 in the revised manuscript.

We deleted the last sentence of introduction in original manuscript, and it was changed into "but also describes a regulatory pattern that CRP-cAMP modulates biofilm maintenance acting as a post-translation regulator". Please see line 126 to line 127 in the revised manuscript.

In addition, I have concerns of how the authors cite or refer the results from CRP-Ecoli in a general way. Sentences 58-66 in the Introduction refer to CRP from *E. coli* and it must be stated as such. CRP proteins from other bacteria have been studied, and some examples are provided by the authors. But the mechanisms and biophysical and structural studies in the first sentences of the introduction refer to the *E. coli* homolog. For instance, others have shown biophysically that binding to promoter sequences is cAMP independent for *M. tuberculosis*. For instance:

a. Stapleton M, Haq I, Hunt DM, Arnvig KB, Artymiuk PJ, Buxton RS, Green J. *Mycobacterium tuberculosis* cAMP receptor protein (Rv3676) differs from the *Escherichia coli* paradigm in its cAMP binding and DNA binding properties and transcription activation properties. *J Biol Chem.* 2010 Mar 5;285(10):7016-27. doi: 10.1074/jbc.M109.047720. Epub 2009 Dec 22. PMID: 20028978; PMCID: PMC2844151.

b. Gárate F, Dokas S, Lanfranco MF, Canavan C, Wang I, Correia JJ, Maillard RA. cAMP is an allosteric modulator of DNA-binding specificity in the cAMP receptor protein from *Mycobacterium tuberculosis*. *J Biol Chem.* 2021 Jan-Jun;296:100480. doi: 10.1016/j.jbc.2021.100480. Epub 2021 Feb 26. PMID: 33640453; PMCID: PMC8026907.

Reply: Both *Shewanella putrefaciens* CN32 and *Escherichia coli* belong to γ -proteobacteria. *E. coli*, as a model strain, has more and relatively comprehensive studies on CRP-cAMP. And we performed heterologous complementation CRP (88% sequence identity with CN32 CRP) of *E. coli* MG1655 in

Δcrp of CN32 can partly restore the biofilm maintenance of Δcrp (Supplementary Fig. 4g). Therefore, we mainly referred to the study of CRP-cAMP in *Escherichia coli* in the original manuscript. According to the suggestion of reviewer 1, we added some updated information of the role of CRP-cAMP in *Mycobacterium tuberculosis* “Although most subsequent studies focused on CRP-cAMP as a transcription factor, CRP-cAMP may play a non-transcriptional regulatory role in some bacteria. For instance, in *Mycobacterium tuberculosis*, CRP may act as a nucleoid associated protein (NAP) to influence the dynamic spatial arrangement of the chromosome in a cAMP-independent manner. However, in γ -proteobacteria, the researches on CRP-cAMP as a non-transcriptional regulator involved in physiological metabolism are limited”. Please see line 71 to line 76 in the revised manuscript. And we added new reference 16, 17, and 18 in the revised manuscript.

I also have concerns on the results from the mutation study using CRP-R84L. The authors claim that this mutant abolishes cAMP activity. Did the authors measure this? I couldn't find it. They cite reference #3 but this is for *E. coli*. While the position seems to be conserved, it is important that the authors show that there is no detectable cAMP binding in order to reach the conclusion that “the ability of CRP to enhance the interaction between BpFD and BpFG is lost in the absence of cAMP”

Reply: Thank you for this critical comment. We provided new experimental results as Supplementary Fig. 3g and Supplementary Fig. 4g in the revised manuscript. We compared the interaction ability of cAMP with CRP and CRP-R84L by biotinylated cAMP pull-down assay (see new added method from line 640 to line 651 of the revised manuscript). The experimental results showed that cAMP could interact with CRP, but the interaction between cAMP and CRP-R84L was not detected (Supplementary Fig. 3g), which is consistent with the conclusions of previous reference (see reference 3 of the revised manuscript). In addition, the heterologous complementation CRP-R83L of *E. coli* MG1655 in Δcrp of CN32 cannot restore the biofilm maintenance of Δcrp (Supplementary Fig. 4g). These results provide new experimental evidences for the conclusion that the ability of CRP to enhance the interaction between BpFD and BpFG is lost in the absence of cAMP.

Lastly, the authors seem to claim in the Discussion that their findings may apply or may be more ubiquitous in bacteria. They state that heterologous complementation can partially restore biofilm maintenance with CRP from *E. coli* in Δcrp but not in $\Delta crp\Delta cya$ but they data is not shown? This is a major claim of this article and the authors must show those results to support their claim that CRP-cAMP may play in biofilm is most bacteria (i.e., ubiquitous).

Reply: This is a valuable suggestion. In the revised manuscript, we added biofilm phenotype of strain $\Delta crpCcrp_{MG1655}$ and $\Delta crp\Delta cyaCcrp_{MG1655}$. Please see **Supplementary Fig. 4g** of the revised manuscript. The results showed that heterologous complementation CRP (88% sequence identity with CN32 CRP) of *E. coli* MG1655 in Δcrp of CN32 can partly restore the biofilm maintenance of Δcrp , but it cannot restore the biofilm biomass in $\Delta crp\Delta cya$, indicating that the homologous CRP in *E. coli* has a function similar to CN32 CRP. Taken together, the above findings may suggest that the regulatory pattern of CRP acting as post-translation regulator is ubiquitous in bacteria.

Altogether, the study is comprehensive and detailed (somewhat difficult to read too given the specialized terminology) but some assays and measurements are needed. But overall the impact and novelty of the work will better suited in a more specialized journal.

Reply: We thank the reviewer #1 for his professional comments and very good suggestions again. According to the suggestion of reviewer 1, we added new experimental results on the interaction analysis of cAMP with CRP and CRP-R84L (**Supplementary Fig. 3g**), and the biofilm phenotype of strain $\Delta crpCcrp_{MG1655}$ and $\Delta crp\Delta cyaCcrp_{MG1655}$ (**Supplementary Fig. 4g**) in the revised manuscript, Please see our response above. Nature Communications has a large number of readers of research related to biofilm and bacterial second messenger. We will be very honored if our articles can be published on Nature Communications after revising according to your suggestions.

Reviewer #2 (Remarks to the Author):

Liu and colleagues present a voluminous amount of work using complementary techniques from molecular genetics, biochemistry and molecular biology supporting a regulatory mechanism whereby c-AMP-CRP interacts with a membrane-bound receptor to affect biofilm formation in microplates. The regulatory mechanism reported in the paper represents a convergence point for cAMP and c-di-GMP signaling. Many reports in the field recognize cross-talk between these second messengers

yet mechanisms explaining the connections remain scant. The discovery that the cAMP-CRP transcription factor post-translationally regulates a receptor complex important for processing an adhesin is highly novel and will be of interest to many in the field. The work appears technically sound with only a couple of controls that appear to be missing (outlined below). However, the work would benefit greatly from editing and a key shortcoming, which is addressable, is a lack of integration with terminology describing the biofilm ecological cycle.

Reply: We thank the reviewer #2 for his professional comments to our manuscript. We highly agree with the viewpoint of reviewer 2, many previous reports recognized the complex cross-talk between different second messengers, however the molecular mechanisms explaining the connections remain scant. In this report, we not only revealed that cAMP and c-di-GMP synergistically regulate biofilm maintenance through the direct interaction of their effectors but also described that CRP-cAMP post-translationally regulates a receptor complex important for processing an adhesin.

According to the suggestion of reviewer 2, we added new data on controls, please see Supplementary Fig. 3d. And we added new information as following: “The relationships between intracellular c-di-GMP level and bacterial lifestyles have been well-established. Specially, a high intracellular c-di-GMP level is associated with biofilm formation, while a low intracellular c-di-GMP level tends to facilitate a planktonic lifestyle. However, several reports demonstrate that some regulators act as additional tool for fine-tuning such an important cellular molecular mechanism by cross-talking with c-di-GMP. Our results reveal that in limited intracellular c-di-GMP level condition, CRP-cAMP plays a determinant function in the regulation of biofilm maintenance in *S. putrefaciens* CN32 (Fig. 1b, c), which underlines the complexity of bacterial second messenger regulation again. In summary, in *S. putrefaciens* CN32, CRP-cAMP acts as not only global transcription factor to regulate physiological metabolism but also post-translation regulator to participate in biofilm maintenance by cross-regulating with second messenger c-di-GMP”. Please see line 353 to line 363 in the revised manuscript.

Major points of constructive feedback

A major constructive criticism for the authors is that the term biofilm “maintenance” is not integrated within the context of the biofilm ecological cycle that is summarized succinctly on lines 79-80.

Reply: Thanks for pointing this out. According to the suggestion of reviewer 2, we integrated “maintenance” in describing the biofilm ecological cycle as following: “The bacterial biofilm developmental process includes four stages: (i) initial attachment, (ii) microcolony formation, (iii) mature biofilm maintenance, and (iv) dispersal”. Please see line 79 to line 80 in the revised manuscript.

At what stage of this ecological cycle is Crp exerting an effect? Is the cAMP-CRP complex actively controlling a process of biofilm dispersion? That’s the case with the Lap system in *Pseudomonas*, which the authors contrast to the BpgAGD system. The Lap system regulates biofilm dispersal in response to phosphate starvation (see <https://www.pnas.org/content/106/9/3461>). With respect to the writing, the work would benefit greatly from better integration with existing terminology and concepts used in the field.

Reply: This is a very interesting question. In this study, we found that CRP-cAMP can support the biofilm maintenance by regulating BpfAGD system. And in CRP-deleted or cAMP-negative strains, biofilm rapidly disperse. The results of this study revealed the role of CRP-cAMP in the biofilm maintenance stage, but it is uncertain about whether CRP-cAMP is directly involved in the process of biofilm dispersion. Thus, in future research, we will continue to focus on the function of CRP-cAMP in the initial attachment stage and dispersion stage of biofilm.

Lap system is the best well-known adhesion protein system, which plays a decisive role in the biofilm developmental process of *Pseudomonas fluorescens* Pf0-1. Lap system participates in regulating different stages of biofilm development by responding to external environmental signals, such as phosphate starvation signal. Moreover, LapD in Lap system can be activated by directly interacting with multiple DGCs and respond to local c-di-GMP signaling. These reported studies on Lap system provide a good reference for the study of adhesion protein systems in other bacteria. BpfAGD system in *S. putrefaciens* CN32 belong to Lap system. However, the research on BpfAGD system is limited. It is uncertain whether the BpfAGD system responds to the extracellular signal of phosphate starvation. We will pay attention to this problem in following research.

With respect to better integration with the biofilm ecological cycle, can the authors provide time-dependent microscopy data to look at what’s happening in their CRP mutant? For example, one

possible interpretation is that cAMP-CRP suppresses nutrient-dependent biofilm dispersion. Another possibility is that the cells die (such is the case with another study looking at biofilm “maintenance,” <https://pubmed.ncbi.nlm.nih.gov/33531388/>). These outcomes could be evaluated using one of several continuous or batch-culture biofilm models amenable to microscopy. This does not need to be a major undertaking to provide this key information.

Reply: This is a very good and professional suggestion. Through genetic and biochemical methods, this study revealed the signal transduction pathway for CRP-cAMP and c-di-GMP synergistically regulating biofilm maintenance. Time dependent microscopy observation can further reveal the biofilm maintenance phenotype of Δcrp mutant, but this does not affect the main conclusions of this study. We are very sorry for that the method of observing biofilm by confocal microscope in our laboratory is not mature at present. We will try our best to observe and analyze the mechanism of CRP-cAMP supporting biofilm maintenance using microscope observation in future research.

A tremendous amount of work has gone into generating deletion mutants and plasmids for expressing multiple genes simultaneously; however, the vector controls used in the complementation analyse **Reply:** Thanks for this valuable suggestion. In this study, the gene deletion mutants were constructed by in-frame deletion methods. The plasmid pBBR1MCS-2 was used to construct the complementation strains. According to the suggestion of reviewer 2, we provided the new data that the plasmid pBBR1MCS-2 did not affect the biofilm phenotype of *S. putrefaciens* CN32 in the **Supplementary Fig. 3d** of the revised manuscript.

Lines 211-289. It is very difficult to follow the logic in this section because of the way it is written. I’m also unsure whether all this work is necessary for this one manuscript and whether communication might be better if some of the information and data were removed. It’s uncanny that CRP regulates the transcription of so many genes for DGCs and PDEs! It would be great if the authors could home in on just the results that lead the reader to conclude that cAMP-CRP affects biofilm formation independently of the regulation of all these genes. Also, it isn’t clear why a handful of DGC proteins would need to be produced, purified and verified for activity because of the pleiotropic effects Crp has on gene regulation. Can the authors trim down the data to the key dataset(s) and simplify the presentation?

Reply: Thanks for this professional comment. At present, the research on the molecular mechanism of biofilm formation in *S. putrefaciens* CN32 is limited. BpfAGD system and c-di-GMP signaling are two known key factors regulating biofilm formation in *S. putrefaciens* CN32. The second part of the result indicates that CRP-cAMP regulates biofilm maintenance independent of its regulation of *bpfA* transcription. Thus, we next considered whether CRP-cAMP controls biofilm maintenance through regulating the intracellular c-di-GMP level. Most bacteria have more than one DGC/PDE, the quantity of which is associated with the complexity of the living environment of bacteria. Although most bacteria have multiple DGCs/PDEs, only a few DGCs/PDEs influence biofilm development at a defined time period. Therefore, the regulation of c-di-GMP on bacterial biofilm is very complex. We discussed this information in line 85 to line 88 of the revised manuscript.

We identified 47 DGC or PDE genes in the genome of *S. putrefaciens* CN32. To screen biofilm maintenance-related *dgc/pde* genes whose transcriptions are regulated by CRP-cAMP, the transcription levels of all 47 *dgc/pde* genes were compared in WT and Δcrp at 30 h. Although CRP-cAMP regulates the transcription of 41 of the *dgc/pde* genes (**Supplementary Table 1**), only two *dgc* genes *Sputcn32_1291* and *Sputcn32_3328* were identified to influence biofilm biomass at 30 h. In order to further verify the DGC activities of the two enzymes, both proteins were purified, and their enzyme activities were detected by HPLC. However, by comparing the intracellular c-di-GMP content, biofilm phenotype and the intracellular protein content of *Sputcn32_1291* and *Sputcn32_3328*, we found some contradictory results, which led us to find that CRP-cAMP regulates biofilm maintenance independent of its regulation of the intracellular c-di-GMP concentration. We agree with the reviewer's suggestion that some unnecessary information were removed from this part result. We have rewritten this part. Please see line 226 to line 247 in the revised manuscript.

CRP-cAMP is a typical global transcription factor, which regulate the transcription of a large number of genes in different bacteria. For examples, CRP-cAMP can regulate the transcription of 7% genes in *Escherichia coli*, 6% genes in *Vibrio cholerae* and more than 200 genes in *Yersinia pestis*. In this study, we also found that CRP-cAMP regulates the transcription of a large number of *dgc/pde* genes, but only two DGCs affect the intracellular c-di-GMP concentration during biofilm maintenance. And our further study found that CRP-cAMP regulates biofilm maintenance independent of its regulation of the intracellular c-di-GMP concentration. This also led us to find that CRP-cAMP modulates an adhesion system by directly interacting with a c-di-GMP effector BpFD.

In addition, according to the suggestion of reviewer 2, the transcription data in Supplementary Fig. 4 of the original manuscript is presented as **Supplementary Table 1** of the revised manuscript.

Minor points

Lines 93-108. A comparison is made between the BpfAGD systems and the *Pseudomonas* Lap system. Are the systems orthologous, or is this comparison simply analogous? Could the authors provide more detail about the molecular functioning of the systems and clarify, please?

Reply: Thanks for this question. Although adhesion protein systems have been identified in many bacteria, Lap system of *Pseudomonas fluorescens* Pf0-1 is the best-investigated one. A large number of research literatures have reported the regulation mode of Lap system and the extracellular signals responded by Lap system, which have revealed how Lap system responds to local and global c-di-GMP signaling in *P. fluorescens* Pf0-1. In *Shewanella putrefaciens* CN32, BpfAGD system belong to Lap system and is also a very important adhesion system involved in biofilm, but the research on BpfAGD system is limited. It is known that, similar to LapD of Lap system, BpfD can respond to intracellular global c-di-GMP signal. A high intracellular c-di-GMP level activates the c-di-GMP effector BpfD to bind to and sequester the periplasmic protease BpfG, which prevents BpfA being processed and results in biofilm formation. When intracellular c-di-GMP level is low, BpfD cannot sequester BpfG, leaving BpfG free to process and release BpfA from the cell surface, leading to planktonic mode. We have discussed this information in line 103 to line 109 of the revised manuscript. Reference 42 of the revised manuscript compared the similarity of LapA homologous adhesion proteins in many bacteria, including BpfA protein. They found that although LapA homologs exist in many bacteria and play similar functions, their sequence identity is very low. Through sequence alignment, we found that the domain composition and arrangement of BpfD are similar to LapD, but their sequence identity is only 25%. The domain composition and arrangement of BpfG are also similar to LapG, and their sequence identity is 40%. In addition, studies on the extracellular signal that BpfAGD system respond to and whether BpfAGD system can respond to local c-di-GMP signaling through interaction with DGC/PDE are still limited. Thus, in future research, we will perform more studies on BpfAGD system to further reveal the important role of adhesion system in bacterial biofilm development by comparing the similarities and differences between BpfAGD and Lap systems.

As it pertains to this point, Supplementary Figure 1 and Figure 6 are redundant. The two could be combined. I would suggest presenting the model as Figure 1 in the paper because basic information about the BpfAGD system is critical for understanding the content in the paper. Presenting a revised model up front will allow the readers to better follow along, even if the cAMP-CRP interaction with BpfD, which is the major contribution forthcoming from the present manuscript, is presented first. Also – aren't there 4 outcomes from the model and not just two (because there are 2 second messengers at play, both of which might be high or low)?

Reply: Combining figure 6 and figure 1 is a good suggestion. In the revised manuscript, we provided a new regulation pattern including four conditions. Please see Figure 1 of the revised manuscript. And we added new Figure caption for Figure 1.

In addition, we added new information for this regulatory pattern as following: “The relationships between intracellular c-di-GMP level and bacterial lifestyles have been well-established. Specially, a high intracellular c-di-GMP level is associated with biofilm formation, while a low intracellular c-di-GMP level tends to facilitate a planktonic lifestyle. However, several recent reports demonstrate that some regulators act as additional tool for fine-tuning such an important cellular molecular mechanism by cross-talking with c-di-GMP. Our results reveal that in limited intracellular c-di-GMP level condition, CRP-cAMP plays a determinant function in the regulation of biofilm maintenance in *S. putrefaciens* CN32 (Fig. 1b, c), which underlines the complexity of bacterial second messenger regulation again. In summary, in *S. putrefaciens* CN32, CRP-cAMP acts as not only global transcription factor to regulate physiological metabolism but also post-translation regulator to participate in biofilm maintenance by cross-regulating with second messenger c-di-GMP”. Please see line 353 to line 363 in the revised manuscript.

Lines 116 and 371 (and elsewhere). “...bacterial bistable transition.” Bistability refers to a very specific type of gene expression, and this term is not used correctly in the context of this paper. Do the authors mean “the transition between motile planktonic and sessile biofilm lifestyles in many bacteria?”

Reply: Thanks for this suggestion. We have changed all “bacterial bistable transition” into “the transition between motile planktonic and sessile biofilm lifestyles”. Please see line 117 to line 118, and line 368 in the revised manuscript.

Lines 132-134. If the bacteria are still growing, they are not in stationary phase. Do the authors mean that the *crp* mutant grew to a higher cell density than the other strains?

Reply: Thanks for this suggestion. We have changed the sentence “but slightly faster at the stationary phase (after 12 h)” into “but Δcrp grew to a slightly higher cell density than the other strains after 12 h”. Please see line 135 to line 136 in the revised manuscript.

Line 136 (and elsewhere). Remove “obvious” and replace with “statistically significant.”

Reply: Thanks for this suggestion. We have replaced all the word “obvious” or “obviously” with “statistically significant” or “significantly”. Please see line 138, 184, 258, 260, and 273 of the revised manuscript.

Line 142. “tertiary-knockout strain.” I recommend using the language of geneticists, which would be to state “a strain lacking all 3 adenylate cyclases.” Similarly, “tertiary-complementation strain” should be a “strain bearing a plasmid encoding all 3 adenylate cyclases.” Some additional details added to the results section would better help the reader to understand how these constructs were built.

Reply: Thanks for this good suggestion. We have replaced “tertiary-knockout strain” with “a mutant lacking all 3 adenylate cyclases”, and replaced “tertiary-complementation strain” with “its complementation strain bearing a plasmid encoding all 3 adenylate cyclases”. Please see line 144 to line 146 of the revised manuscript.

Lines 172-174 (figure 2B). How do the authors know that cAMP isn’t carried over from media to the nucleotide extractions? I’m not sure if that matters – but wording around presentation of the data could be more careful.

Reply: Thanks for this question. The biofilm analysis of *S. putrefaciens* CN32 was carried out in modified M1 defined minimal medium (MM1), which is a chemically synthetic medium with clear

components without cAMP. Please see line 452 to line 455 of the revised manuscript. In addition, when we detected the intracellular cAMP concentration, the bacterial cells were washed twice with cold PBS buffer to remove the residue of culture medium after cell collection by centrifugation. Please see line 580 to line 583 of the revised manuscript.

Line 200 (Figure 2J). There are multiple numbers that appear in Fig. 2J with respect to genotypes and these are not explained sufficiently in the text when the data are first presented.

Reply: Thanks for pointing this out. In order to avoid providing duplicate data, we put all the biofilm data related to *bpfD* and *bpfG* genes in the same figure, which is more conducive to accurately compare the biofilm phenotypes of different strains. At last, we thank reviewer 2 again for his useful suggestion to our manuscript.

Reviewer #3 (Remarks to the Author):

In this manuscript the authors investigate aspects of the molecular mechanisms of biofilm formation mediated by the cyclic AMP second messenger signaling system in *Shewanella putrefaciens* CN32. They could show that the cyclic AMP receptor protein CRP in complex with its ligand cAMP, a transcriptional regulator, regulates maintenance of biofilm formation in the late phase of growth upon incubation under steady state conditions not only by transcriptional regulation of cyclic di-GMP turnover proteins, but also by direct interaction with the cyclic di-GMP receptor BpfD. BpfD, upon cyclic di-GMP binding and interaction with the cAMP-loaded CRP, prevents the protease BpfG to cleave the large surface protein BpfA in order to dissolve the biofilm.

This work adds a major novel aspect to the complex regulation of biofilm formation in *S. putrefaciens* and the interplay between components of two ubiquitous second messenger signaling systems. Furthermore, the authors show that the molecular mechanisms of regulation of this work potentially extend to other bacterial species, in particular *Escherichia coli*. The work is presented very organized and overall clearly presented. It would be however highly beneficial to strengthen the findings by identification of amino acids in CRP which do not promote interaction with BpfD.

Reply: We thank reviewer 3 for his good suggestions and professional comments to our manuscript. Identification of key functional amino acids in CRP is a good suggestion. At present, we have

identified the key amino acid site binding cAMP in CRP by site-mutation technology in *S. putrefaciens* CN32. However, because CRP and BpFD may have conformation transformations when they bind to their ligands cAMP and c-di-GMP, we have not identified the amino acid sites where CRP interacts with BpFD, but this does not affect the conclusion of this manuscript. In future research, we will focus on finding the amino acid sites where CRP interacts with BpFD.

As *S. putrefaciens* has three adenylate cyclases, it would be relevant to know whether all three enzymes contribute equally to biofilm maintenance in the stationary phase, or whether one of those enzymes has a dominant role. Does any of those proteins contain an N-terminal signaling domain that allows conclusions about a specific activation in the stationary phase of growth? Perhaps this part of the signaling pathway in combination with the cyclic di-GMP turnover proteins that are regulated by cAMP-CRP can be included in the figure.

Reply: This is a very interesting question. For the influence of three single *cya* gene deletion mutants on biofilm maintenance, CyaA is the main adenylate cyclase to maintain biofilm, and CyaC also partially contributes to maintaining biofilm, but only a small contribution. The deletion of *cyaB* did not affect the biofilm maintenance. In order to construct a cAMP-negative strain, we deleted three *cya* genes at the same time.

Until now, we have not found important signal functional domains at the N-terminal of these Cya proteins, especially for CyaA, which plays a major role in maintaining biofilm. CyaA protein in *S. putrefaciens* CN32 shows only 29% homology with CyaA of *E. coli*. At present, the study on three *cya* genes in *S. putrefaciens* CN32 is limited. Thus, we think this is a good suggestion. We will pay more attention to three *cya* genes in future research.

Specific comments:

l.32: ...the complex cross-regulation of various second messengers has been hardly explored.

Reply: We thank reviewer 3 for polishing the language of our article. These language modification suggestions are very helpful to improve our manuscript. We have changed the sentence “the cross-regulation of various second messengers are very complicated and not well established” into the sentence “the complex cross-regulation of various second messengers has been hardly explored”. Please see line 31 to line 32 in the revised manuscript.

l.33: textbook? ..one of the most well investigated transcription factors.

Reply: We have removed the word “textbook” in the revised manuscript.

l.34: ...act not only as a transcription factor, but also through interaction with an effector protein.

Reply: We have changed the sentence “The cAMP receptor protein (CRP) is ubiquitous and acts as a textbook transcription factor, but there has been no report on the ability of CRP-cAMP to act as a non-transcriptional regulator” into “The cAMP receptor protein (CRP) is a well-characterized global transcription factor, but investigations on CRP-cAMP as a post-translation regulator are limited”. Please see line 32 to line 34 in the revised manuscript.

l.36: ...to regulate proteolytic cleavage of BfpA by BpfG.

Reply: We have changed the sentence “to regulate the BpfAGD adhesion system” into the sentence “to regulate proteolytic cleavage of BfpA by BpfG”, please see line 35 to line 36 in the revised manuscript.

l.125: ...acting not only as a transcriptional regulator by DNA binding, but also acting through protein-protein interactions with a cyclic di-GMP receptor to modulate effector efficiency.

Reply: We have changed the sentence “but also describes for the first time that CRP-cAMP, acting as a non-transcriptional regulator, modulate a non-transcriptional-associated signal transduction pathway” into “but also describes a regulatory pattern that CRP-cAMP modulates biofilm maintenance acting as a post-translation regulator”. Please see line 126 to line 127 in the revised manuscript.

l.154: ...we deleted...

Reply: We have replaced “We next knocked out” by “We deleted” in the line 157 in the revised manuscript.

l.189: As the authors point out later in the manuscript, not necessarily, in particular also as cAMP-CRP as a transcription factor affects transcription of hundreds of genes that might also be involved in the process of surface exposure of BpfA

Reply: Thanks for this comment. This is just a hypothetical possibility. In the revised manuscript, we have changed the sentence “Logically speaking, if CRP-cAMP regulates biofilm maintenance dependent on promoting the transcription of *bpfA*, the biofilm biomass of $\Delta crp/P_{aacC1-bpfA}$ should be higher than that of Δcrp and close to that of WT” into “If CRP-cAMP regulates biofilm maintenance dependent on promoting the transcription of *bpfA*, the biofilm biomass of $\Delta crp/P_{aacC1-bpfA}$ should be higher than that of Δcrp and close to that of WT”. Please see line 191 to line 193 of the revised manuscript.

l.234: Besides the full domain structure, can the authors comment whether and which domains are expected to be functional for 1291 and 3328

Reply: Both protein Sputcn32_1291 and Sputcn32_3328 contain conserved GGDEF domain of diguanylate cyclase (DGC) activity, which is the target domain when we detected their DGC activity. We have discussed this information from line 244 of the revised manuscript.

l.267 and elsewhere: Not really, as cAMP-CRP positively regulates the transcription of the bpfAGD operon.

Reply: This is just a hypothetical possibility. In the revised manuscript, we have changed the sentence “logically speaking, if CRP-cAMP regulated biofilm maintenance by modulating the intracellular c-di-GMP level, the increase in intracellular c-di-GMP concentration of Δcrp to the WT level should restore its biofilm biomass to the WT level” into “if CRP-cAMP regulated biofilm maintenance by modulating the intracellular c-di-GMP level, the increase in intracellular c-di-GMP concentration of Δcrp to the WT level should restore its biofilm biomass to the WT level”. Please see line 254 to line 257 of the revised manuscript.

l.323: ...integration of a 3XFLAG-tag nucleotide sequence 5' of the stop-codon of...

Reply: we have changed the sentence “a 3×FLAG tag was knocked into the C-terminus of BpfD, and a 1×HA tag was knocked into the near C-terminus of BpfG” into “a chromosomal C-terminal

3×FLAG-tag was attached to BpFD, and a chromosomal near C-terminal 1×HA-tag was attached to BpfG”. Please see line 309 to line 312 of the revised manuscript.

1.452: ---by bacteria themselves...

Reply: We have replaced “bacteria-selves” by “by bacteria themselves” in the line 446 in the revised manuscript.

1.495: ...was introduced into...

Reply: We have changed all “was transformed into” by “was introduced into” in the line 471, 558, 563 in the revised manuscript.

1.504: The nucleotide sequences corresponding to tags were inserted upstream of the...

Reply: We have changed the sentence “All tags were knocked into the corresponding location of the labeled gene in the genome” into “The nucleotide sequences encoding tags were inserted into the corresponding location of the target gene in the genome”. Please see line 480 to line 481 in the revised manuscript.

1.516: ..inserted or integrated...

Reply: We have replaced all “knocked-in” by “inserted” in the line 480, 492, 497, 500, and 532 in the revised manuscript.

References for cyclic di-GMP signaling are highly biased. The authors might want to consider to include references that report the original finding that biofilm formation is positively regulated by cyclic di-GMP in a number of organisms.

Reply: *S. putrefaciens*, *E. coli* and *Pseudomonas fluorescens* all belong to γ -proteobacteria, and *S. putrefaciens* CN32 and *P. fluorescens* Pf0-1 have similar BpfAGD system and Lap system. Therefore, the existing studies on *P. fluorescens* and *E. coli* provide a good reference for our research. According to the suggestion of reviewer 3, we updated some references on the regulation of c-di-GMP to biofilm in *Vibrio cholerae*, *Clostridium difficile*, *Listeria monocytogenes*, *Bacillus subtilis* in the revised manuscript. Please see references 27, 33, 40, and 41 in the revised manuscript.

Reviewers' Comments:

Reviewer #1:

Remarks to the Author:

The authors have adequately address the concerns of this reviewer. The significance of the work is better stated, the changes in wording in some sections helps the reader (especially a non-expert) to better understand the work in a larger context, and the few control experiments requested were executed. I don't have any other comments on this manuscript.

Reviewer #2:

Remarks to the Author:

Liu and colleagues have made a genuine attempt to address the reviewer comments. The addition of control data is appreciated. However, the text of the manuscript, while improved in some places, has declined in others.

There may have been a miscommunication with respect to my past feedback – and my apologies if there has been a lack of clarity in my comments. The previous major point of feedback was that the interpretation of the data in the manuscript was not integrated with the canonical biofilm ecological cycle. For a very recent review of the ecological cycle, I refer the authors to “Biofilm Dispersion” by Rumbaugh and Sauer (2020) in Nature Reviews Microbiology (<https://www.nature.com/articles/s41579-020-0385-0>). This model will be the lens with which many microbiologists look at the present work – for better or for worse – because even if overly generalized, the biofilm ecological model is derived from decades of direct visual observations of biofilms in a variety of systems by microscopy.

Specifically, in the biofilm ecological cycle there is no step referred to as “mature biofilm maintenance” (or “biofilm maintenance”). Rather the step the authors are referring to is termed, “biofilm maturation.” This terminology appeared correctly on lines 79-80 of the original submission but is now incorrect on line 80 of the revised paper. The term “biofilm maintenance” is the one causing confusion. By my estimate, the term “biofilm maintenance” now appears >70 times in the manuscript including the title.

In this respect the authors have not addressed my single greatest concern in the interpretation of the data, which is that the reader can't easily understand what step of the biofilm ecological cycle the CRP-cAMP switch (which is described in elegant detail) is operating. This concern has not yet been resolved. One interpretation is that cAMP and c-di-GMP converge at a checkpoint that regulates biofilm dispersion. Collectively, these concerns/queries were also the basis for requesting the analysis of the *Shewanella* biofilms by microscopy.

Overall, I hope this feedback is helpful to the authors.

Reviewer #3:

Remarks to the Author:

This is the revised version of a manuscript previously submitted to Nature Communication. The manuscript has been greatly improved, however one of my major concerns is still that the most important results that lead to the conclusion of an additional role of cyclic AMP-CRP beyond its role as a transcription factor and which are also making up the major message of this work are provided in short basically from line 287.

In the opinion of this reviewer, additional experimentation is required to confirm the role of cyclic AMP-CRP in interaction with BpFD and the strengthening of the interaction between BpFD and BpFG. In addition, the additional experiment intended to show a similar role of CRP of *Shewanella oneidensis* in *E. coli* need to include the CRPR84L mutant which does not bind cyclic AMP. Does *E. coli* have a system homologous to the BpF system?

Additional comments

I.86: genome size and life style (environment)

I.192: transcription and production

I.198: there are at least two modes how cyclic AMP-CRP regulate bpfA, one is on the transcriptional level.

I.224: It cannot be excluded that expression of dgcQ activates an alternative biofilm pathway. Although the association between high cyclic di-GMP levels and positive regulation of biofilm formation is strong, it is not absolute due to spatial effects etc. This has been shown in several screening studies.

Point to point response to REVIEWER COMMENTS:

Reviewer #1 (Remarks to the Author):

The authors have adequately address the concerns of this reviewer. The significance of the work is better stated, the changes in wording in some sections helps the reader (especially a non-expert) to better understand the work in a larger context, and the few control experiments requested were executed. I don't have any other comments on this manuscript.

Reply: We thank Reviewer #1 for his satisfaction to our revised manuscript.

Reviewer #2 (Remarks to the Author):

Liu and colleagues have made a genuine attempt to address the reviewer comments. The addition of control data is appreciated. However, the text of the manuscript, while improved in some places, has declined in others.

There may have been a miscommunication with respect to my past feedback – and my apologies if there has been a lack of clarity in my comments. The previous major point of feedback was that the interpretation of the data in the manuscript was not integrated with the canonical biofilm ecological cycle. For a very recent review of the ecological cycle, I refer the authors to “Biofilm Dispersion” by Rumbaugh and Sauer (2020) in Nature Reviews Microbiology (<https://www.nature.com/articles/s41579-020-0385-0>). This model will be the lens with which many microbiologists look at the present work – for better or for worse – because even if overly generalized, the biofilm ecological model is derived from decades of direct visual observations of biofilms in a variety of systems by microscopy. Specifically, in the biofilm ecological cycle there is no step referred to as “mature biofilm maintenance” (or “biofilm maintenance”). Rather the step the authors are referring to is termed, “biofilm maturation.” This terminology appeared correctly on lines 79-80 of the original submission but is now incorrect on line 80 of the revised paper. The term “biofilm maintenance” is the one causing confusion. By my estimate, the term “biofilm maintenance” now appears >70 times in the manuscript including the title.

Reply: We thank Reviewer #2 for his new professional comments and useful suggestions to our revised manuscript. And we are sorry for that we didn't deeply understand your past feedback. We agree with your current comment, “biofilm maturation” is a more professional term for describing the canonical biofilm ecological cycle, which was mentioned in the literature (reference 67 of our

manuscript) recommended by Reviewer #2. The article published by Professor O'Toole on mBio in 2021 (reference 24 of our manuscript) used the term "biofilm maintenance". The word "biofilm maintenance" refers to **the process by which existing mature biofilms regulate themselves to persist on a surface**, thus we believe that the term "biofilm maintenance" can better reflect the role of CRP-cAMP in maintaining the mature biofilm before dispersion.

We carefully considered this question raised by Reviewer #2 and strongly agreed with this suggestion. In the updated manuscript, we changed it to "biofilm maturation" when we describe the canonical biofilm ecological cycle (please see line 80). Subsequently, we clarify the meaning of "biofilm maintenance" as **the process by which existing mature biofilms regulate themselves to persist on a surface before dispersion** in line 80 to line 84 of the introduction section. This detailed description of "biofilm maintenance" can help readers understand what biofilm stage CRP-cAMP plays a role.

In this respect the authors have not addressed my single greatest concern in the interpretation of the data, which is that the reader can't easily understand what step of the biofilm ecological cycle the CRP-cAMP switch (which is described in elegant detail) is operating. This concern has not yet been resolved. One interpretation is that cAMP and c-di-GMP converge at a checkpoint that regulates biofilm dispersion. Collectively, these concerns/queries were also the basis for requesting the analysis of the *Shewanella* biofilms by microscopy. Overall, I hope this feedback is helpful to the authors.

Reply: Thanks for this valuable suggestion. To make the readers better understand this question, we provided a time-dependent quantitative analysis of biofilm development by biofilm biomass as updated Fig. 2a. It can be seen that the biofilm biomass of WT and Δcrp was similar in the early stage (12, 18 h). After 24 h, the biofilm of Δcrp entered the dispersion stage, while the WT biofilm maintained the mature biofilm state. As the difference in the biofilm biomass of WT and Δcrp was the most obvious at 30 h, the time point of 30 h was selected to study the regulation mechanism of CRP-cAMP on biofilm. In fact, we speculate that the mechanism of CRP-cAMP regulating biofilm ecological cycle is very complex, because through the western experiment, we found that the intracellular CRP content at the protein level was significantly higher in the mature biofilm stage (30 h) than that in the early stage (12 h). Therefore, we speculate that the increase of intracellular CRP

content in the biofilm mature stage may be the reason for the interaction of CRP with BpfD to support biofilm. However, this is our speculation and the experimental evidence is not sufficient. Abundant of experiments are needed to compare CRP regulation of biofilms in early (12 h) and late stages (30 h), so we do not discuss this result in this manuscript. In this report, we focused on the underlying mechanism that CRP regulates biofilm maintenance, which will benefit to the future researches about how CRP participate in biofilm ecological cycle. In future study, we will further investigate how CRP and c-di-GMP participate in the whole biofilm ecological cycle. Altogether, our results of this manuscript suggested that cAMP and c-di-GMP play a synergistic role in maintaining mature biofilm in the biofilm maturation stage. Furthermore, we clarify the term "biofilm maintenance" as the process by which existing mature biofilms regulate themselves to persist on a surface before dispersion in line 80 to line 84 of the introduction section, which can help readers understand that cAMP and c-di-GMP play a synergistic role in the biofilm maturation stage. We thank the professional and valuable comments of Reviewer #2 to help us improve our manuscript again.

Reviewer #3 (Remarks to the Author):

This is the revised version of a manuscript previously submitted to Nature Communication.

The manuscript has been greatly improved, however one of my major concerns is still that the most important results that lead to the conclusion of an additional role of cyclic AMP-CRP beyond its role as a transcription factor and which are also making up the major message of this work are provided in short basically from line 287.

In the opinion of this reviewer, additional experimentation is required to confirm the role of cyclic AMP-CRP in interaction with BpfD and the strengthening of the interaction between BpfD and BpfG.

Reply: We thank Reviewer #3 for his detailed and careful comments on our manuscript again. The parts before and after line 287 (line 292 of updated manuscript) are very relevant, and the important findings after line 287 are based on the phenotypic analysis and biochemical experiments before line 287.

For the interaction of cAMP-CRP with BpfD, we provide at least four direct results to support this conclusion: 1. the biofilm phenotype of Δcrp is similar to that of $\Delta 3328\Delta 1291$, indicating that

similar to c-di-GMP, CRP regulates the biofilm phenotype through BpfD (Fig. 3j); 2. co-IP result (Fig. 5e); 3. GST-pull down result (Fig. 5f); 4. microscale thermophoresis (MST) experiment (Fig. 5g).

For the role of cyclic AMP-CRP in the strengthening of the interaction between BpfD and BpfG, we also provide at least four direct results to support this conclusion: 1. the similar biofilm phenotype of Δcrp and Δcya (Fig. 2d and 2f) and the biochemical result (Supplementary Fig. 3g and 4a-c) indicated that cAMP directly interact CRP and the CRP-cAMP complex regulate biofilm phenotype; 2. co-IP result (Fig. 5d and 6c); 3. western blot experiment (Fig. 3k), because the interaction strength between BpfD and BpfG directly determine the BpfA amount on the cell surface; 4. Δcrp and $\Delta 3328\Delta 1291$ showed similar defective biofilm phenotype, and both $\Delta crp\Delta bpfG$ and $\Delta 3328\Delta 1291\Delta bpfG$ restored the biofilm phenotype (Fig. 3j). Thus, we believe that these experimental evidences are sufficient to support these two conclusions.

In addition, the additional experiment intended to show a similar role of CRP of *Shewanella oneidensis* in *E. coli* need to include the CRPR84L mutant which does not bind cyclic AMP. Does *E. coli* have a system homologous to the Bpf system?

Reply: *E. coli* is a model strain to study biofilm, but until now, Lap/Bpf homologous system has not been reported in *E. coli*. And the review paper published by Professor O'Toole on Annual Review of Microbiology in 2020 (reference 42 of our manuscript) summarized the strains containing Lap or homologous Bpf system, without mentioning *E. coli*. It may be very interesting to study the function of CRP and CRPR84L of *S. putrefaciens* CN32 (we think Reviewer #3 want to say *S. putrefaciens*, not *Shewanella oneidensis*) in *E. coli*, but the biofilm formation mechanism of *E. coli* is obviously different from that of *S. putrefaciens* CN32, and *E. coli* does not contain Bpf homologous system, which limit our analysis of the function of CRP and CRPR84L of *S. putrefaciens* in *E. coli*.

Additional comments

1.86: genome size and life style (environment)

Reply: Thanks for this comment. The published references suggest that the correlation between the genome size and DGC/PDE quantity is not obvious in bacteria. References 31-33 cited in this manuscript show that the DGC/PDE quantity is associated with the complexity of the habitat of

bacteria. To make the readers better understand this information, we changed “living environment” to “habitat”, and we added the following sentence “For example, free-living microbiology tend to have more DGC/PDE than obligate pathogenic bacteria”. Please see line 90 to line 91.

1.192: transcription and production

Reply: Thanks for this suggestion. This study found that the transcription of *bpfA* gene and the total intracellular BpfA protein were not the decisive reasons for influencing biofilm maintenance of *S. putrefaciens* CN32, but BpfA localized on the cell surface was the key to the biofilm maintenance. We have carefully considered your suggestion. We are worried about that adding the word “production” here will cause the confusion of readers, because this will make readers unable to distinguish between the total amount of intracellular BpfA protein and the amount of BpfA localized on the cell surface.

1.198: there are at least two modes how cyclic AMP-CRP regulate *bpfA*, one is on the transcriptional level.

Reply: Thanks for this comment. In this section, we focused on investigating how CRP-cAMP regulates biofilm maintenance. Indeed, as your comment, we found CRP-cAMP directly controls the transcription of *bpfA* using qRT-PCR and EMSA experiments. We have described this result in the original manuscript. Please see line 188 to line 191. However, our following experiments showed that the CRP-cAMP regulating the transcription of *bpfA* and the total intracellular BpfA protein was not the decisive reasons for the effect of CRP-cAMP on biofilm maintenance. We have described this result in the original manuscript. Please see line 192 to line 202. Thus, we concluded that CRP-cAMP regulates the biofilm maintenance independent of its regulation of *bpfA* transcription. Subsequently, our experiments showed that another mode of CRP regulating BpfA localization on the cell surface, which is the main reason for CRP-cAMP supporting biofilm maintenance. We have described this result in the original manuscript. Please see line 205 to line 214.

1.224: It cannot be excluded that expression of *dgcQ* activates an alternative biofilm pathway. Although the association between high cyclic di-GMP levels and positive regulation of biofilm

formation is strong, it is not absolute due to spatial effects etc. This has been shown in several screening studies.

Reply: Thanks for pointing this out. Indeed, many studies have shown that DGC/PDE not only regulate biofilm through regulating global c-di-GMP levels, but also regulate alternative biofilm pathways by spatial effects (local c-di-GMP signaling). The review paper (reference 34 cited in this manuscript) published by Professor Regine Hengge on Trends in Microbiology in 2021 summarized three criteria to distinguish high-specificity local and global c-di-GMP signaling, one of which is whether DGC/PDE can greatly change the total intracellular c-di-GMP concentration. The remarkable feature of DGC/PDE participating in the spatial regulation (local c-di-GMP signaling) is that they hardly affect the total intracellular c-di-GMP concentration, which has been described in reference 34 as "a specific phenotype identified by knocking out a particular DGC or PDE in the absence of consequential changes in cellular c-di-GMP levels is a strong argument for local signaling". The research paper published by Professor Urs Jenal on Cell (doi: 10.1016/j.cell.2010.01.018) indicated that DgcQ can significantly increase intracellular c-di-GMP concentration in its source bacterium *E. coli*. Our study found that the expression of *E. coli* DgcQ in *S. putrefaciens* CN32 can significantly increase intracellular global c-di-GMP level (Fig. 4a). Thus, DgcQ regulated the biofilm through regulating the intracellular global c-di-GMP level, rather than spatial effects.

More important, it is well known that in the BpfAGD system of *Shewanella*, high c-di-GMP level promote the biofilm formation through retaining BpfA on the cell surface. In *S. putrefaciens* CN32, BpfA protein plays a decisive role in the biofilm formation, and the $\Delta bpfA$ mutant cannot form biofilm (Fig. 3a). Our study showed that the biofilm phenotype of $\Delta bpfA/pdgcQ$ is similar to that of $\Delta bpfA$ (Fig. 5a). In other words, once BpfA is deleted, the biofilm cannot be restored by the expression of DgcQ. This further confirmed that DgcQ regulated biofilm through c-di-GMP controlled BpfAGD system, rather than through other alternative biofilm pathways. Moreover, another experiment of this study can also relieve the worry of Reviewer #3. In addition to heterologous expression of *dgcQ* of *E. coli*, we also performed endogenous overexpression of Sputcn32_3328 and Sputcn32_1291 (DGCs of *S. putrefaciens* CN32). Heterologous expression of *dgcQ* and O1291O3328 showed a similar effect on the biofilm phenotype (Fig. 5a), which further confirmed that DgcQ regulated biofilm only by regulating the total intracellular c-di-GMP content.

At last, we thank the Reviewer #3 for his very detailed and careful comments to our manuscript again.

Reviewers' Comments:

Reviewer #2:

Remarks to the Author:

The authors have done a great job addressing my concerns, and should be commended for the effort put into clarifying the written narrative of the paper and providing the extra data requested by the reviewers, including me. The the robust biochemical and genetic studies appearing in this manuscript, including the extra experiments in the supplementary information, are appreciated!